# A3D: Does Diffusion Dream about 3D Alignment?

**Savva Ignatyev**[*1] **Nina Konovalova**[*2] **Daniil Selikhanovych**[1] **Oleg Voynov**[1,2]
**Nikolay Patakin**[2] **Ilya Olkov**[1] **Dmitry Senushkin**[2] **Alexey Artemov**[3]
**Anton Konushin**[2] **Alexander Filippov**[4] **Peter Wonka**[5] **Evgeny Burnaev**[1,2]
[1]Skoltech, Russia   [2]AIRI, Russia   [3]Medida AI, Israel
[4]AI Foundation and Algorithm Lab, Russia   [5]KAUST, Saudi Arabia
*Savva Ignatyev and Nina Konovalova contributed equally
Corresponding author: Savva Ignatyev (e-mail: savva.ignatyev@skoltech.ru)

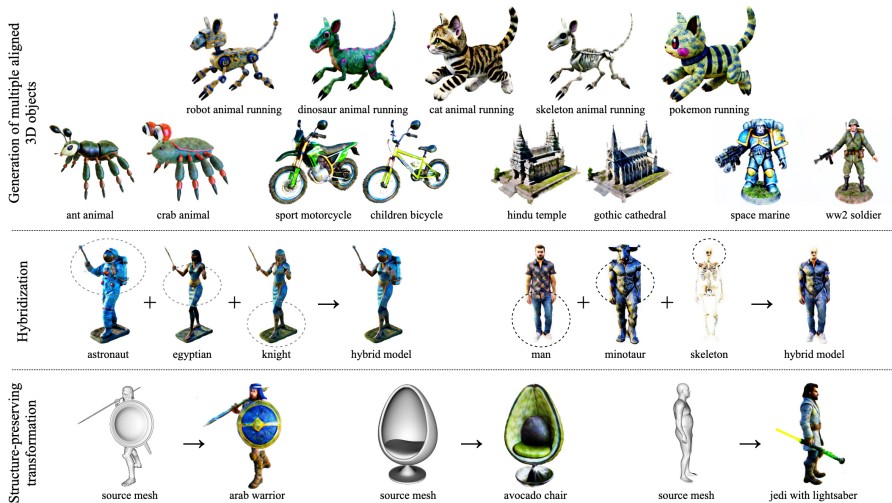

Figure 1: Our method A3D enables conditioning text-to-3D generation process on a set of text prompts to jointly generate a set of 3D objects with a shared structure (top). This enables a user to make "hybrids" combined of different parts from multiple aligned objects (middle), or perform text-driven structure-preserving transformation of an input 3D model (bottom).

## Abstract

We tackle the problem of text-driven 3D generation from a geometry alignment perspective. Given a set of text prompts, we aim to generate a collection of objects with semantically corresponding parts aligned across them. Recent methods based on Score Distillation have succeeded in distilling the knowledge from 2D diffusion models to high-quality representations of the 3D objects. These methods handle multiple text queries separately, and therefore the resulting objects have a high variability in object pose and structure. However, in some applications, such as 3D asset design, it may be desirable to obtain a set of objects aligned with each other. In order to achieve the alignment of the corresponding parts of the generated objects, we propose to embed these objects into a common latent space and optimize the continuous transitions between these objects. We enforce two kinds of properties of these transitions: smoothness of the transition and plausibility of the intermediate objects along the transition. We demonstrate that both of these properties are essential for good alignment. We provide several practical scenarios that benefit from alignment between the objects, including 3D editing and object hybridization, and experimentally demonstrate the effectiveness of our method. voyleg.github.io/a3d

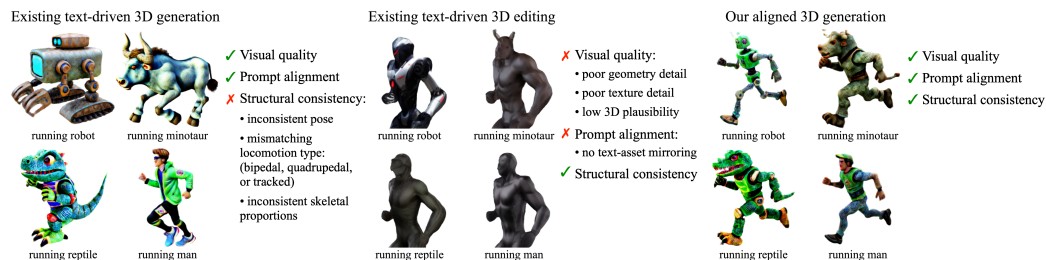

Figure 2: Collections of objects generated with existing text-to-3D methods lack structural consistency (left, (Shi et al., 2024)). Shapes obtained with existing text-driven 3D editing methods lack text-to-asset alignment and visual quality (middle, (Chen et al., 2024a)). In contrast, our method enables the generation of structurally coherent, text-aligned assets with high visual quality (right).

# 1  INTRODUCTION

Creating high-quality 3D assets is a time- and labor-intensive process, so even experienced 3D artists commonly break it down into manageable steps. Prior to shaping an asset, an artist might conceptualize its design structure to capture geometric proportions and spatial relationships of its semantically meaningful parts. Then, a series of detailed 3D design instances (*e.g.*, high-resolution geometry and textures) consistent with an established structure can be produced. Recent 3D generation research (Poole et al., 2023; Qiu et al., 2024; Shi et al., 2024) promises to significantly reduce the effort related to manual production of such high-resolution textured shapes, replacing it by an automated AI-based step controlled with natural language. A text-to-3D generation pipeline could potentially be utilized to produce a collection of structurally aligned 3D objects, consisting of a common set of semantic parts and sharing their structure, *e.g.*, the pose or the arrangement of semantic parts.

However, existing 3D generation approaches synthesize objects independently and fail to maintain structural alignment across them (Figure 2, left). One may attempt to enforce the alignment in a series of 3D objects by generating an initial one with a text-to-3D pipeline and obtaining the others with text-driven 3D editing methods (Haque et al., 2023; Chen et al., 2024a). Unfortunately, the latter struggle with the visual quality and sometimes fail to perform the necessary edits appropriately, resulting in a low degree of the alignment with the text prompt (Figure 2, middle).

To address the limitations of existing approaches, we propose *A3D*, a method for jointly generating collections of structurally aligned objects from a collection of respective text prompts. The idea of our method is to embed a set of 3D objects and transitions between them into a shared latent space and enforce smoothness and plausibility of these transitions. We take inspiration from the transition trajectory regularization proposed for 2D GANs (Karras et al., 2020, Sec. 3.2). We represent each set of 3D objects and the transitions between them with a single Neural Radiance Field (NeRF) (Mildenhall et al., 2020) and train it with a text-to-image denoising diffusion model via Score Distillation Sampling (SDS) (Poole et al., 2023) to simultaneously correspond to the set of text prompts and enforce the plausibility of the transitions between the objects.

Our method is naturally suited for several different scenarios that require control over the structure of the generated objects. (1) Generation of multiple structurally aligned 3D objects (Figure 1, top) enables artists to choose an appropriate 3D asset among a variety of generations, replace 3D objects within existing scenes, or transfer animations across distinct objects. (2) Combining parts of different objects into a "hybrid" (Figure 1, middle) allows adjusting constituent elements without affecting the overall structure of the asset. (3) Structure-preserving transformations of a 3D object (Figure 1, bottom), let a user design a simplified 3D shape with a particular pose and let the automatic generation process fill in complex geometric details and texture while preserving the structure.

Sets of objects generated using our method exhibit a high degree of structural alignment and high visual quality scores, outperforming those obtained with state-of-the-art alternatives. Our method is easily adopted for the structure-preserving transformation, performing on par with specialized text-driven 3D editing methods. Further, it is effective in combination with different text-to-3D generation frameworks. Overall, our work advances the state of the art in text-driven 3D generation and opens up new possibilities for applications requiring the generation of structurally aligned objects.

## 2 RELATED WORK

### 2.1 TEXT-DRIVEN 3D ASSET GENERATION

Collecting large, high-quality, diverse 3D datasets poses significant challenges, so 3D generation approaches predominantly leverage 2D priors for training. DreamFusion (Poole et al., 2023) introduced Score Distillation Sampling (SDS) that enables training Neural Radiance Fields (NeRFs) (Mildenhall et al., 2020) with the guidance of pre-trained 2D diffusion models. Subsequent research has refined this methodology to improve the quality and speed of 3D generation. Magic3D (Lin et al., 2023) uses a coarse-to-fine optimization strategy to increase the speed and resolution. Fantasia3D (Chen et al., 2023a) disentangles the geometry and texture training. Several works enhance realism, detail, and optimization speed by utilizing adversarial training (Chen et al., 2024d), 3D-view conditioned diffusion models (Liu et al., 2023; Shi et al., 2023a; 2024; Liu et al., 2024; Ye et al., 2024; Seo et al., 2024), and Gaussian splatting-based models (Tang et al., 2024b; Yi et al., 2024). All these works focus on independent optimization for distinct prompts, resulting in generation of collections of objects that lack structural alignment, as we show in Figure 2 and in our ablation study. This misalignment issue persists even in amortized frameworks, where a single generative model is trained to handle multiple prompts (Tang et al., 2024a; Jun & Nichol, 2023; Hong et al., 2024; Siddiqui et al., 2024; Ma et al., 2024). Due to the mode-seeking nature of SDS (Poole et al., 2023), these frameworks often produce misaligned objects with the structure sensitive to subtle variations in the prompt, increasing inconsistency across generated outputs. Unlike the described methods that optimize 3D objects independently or amortized models trained on large-scale datasets, our method optimizes a small set of objects jointly, allowing us to achieve structural consistency between objects.

### 2.2 TEXT-DRIVEN 3D ASSET EDITING

One straightforward way to produce a collection of aligned 3D objects is to generate an initial object using a text-to-3D pipeline and subsequently modify this object via text-driven 3D editing. Several methods have been proposed to manipulate NeRF-based scene representations using text as guidance (Haque et al., 2023; Park et al., 2024; Bao et al., 2023; Zhuang et al., 2023). DreamBooth3D (Raj et al., 2023) and Magic3D (Lin et al., 2023) provide the capability to edit personalized objects while leveraging the underlying 3D structure. FocalDreamer (Li et al., 2024), Progressive3D (Cheng et al., 2024), and Vox-E (Sella et al., 2023) confine the effect of modifications to specific parts of the object, thus enhancing control of the editing process. Fantasia3D (Chen et al., 2023a) and DreamMesh (Yang et al., 2024) focus on global transformations of one object into another, iteratively optimizing a 3D model to align with a text prompt via SDS. Iterative optimization with SDS does not guarantee preservation of the structure of the transformed object, so several techniques were proposed to improve it. Coin3D (Dong et al., 2024) refines geometric primitives into high-quality assets by imposing deformation constraints through input masks. GaussianDreamer (Yi et al., 2024) and LucidDreamer (Liang et al., 2024) show text-driven editing capabilities for Gaussian splats, which they initialize using a separate pipeline and fine-tune with the help of a diffusion model. Haque et al. (2023) and Palandra et al. (2024) use the SDS loss in combination with a pre-trained 2D image editing network InstructPix2Pix (Brooks et al., 2023). MVEdit (Chen et al., 2024a) goes one step further by avoiding SDS and proposes a special mechanism that coordinates 2D edits from different viewpoints. Although some of these methods allow obtaining sets of the aligned objects sequentially, the editing process is constrained by the configuration of the initially generated object. This limits the visual quality of the generated sets of objects, as we show in Figure 2 and in our experiments. In contrast, our method optimizes the whole transition trajectory between the objects, and produces both structurally consistent and high-quality results. Additionally, our method is easily adapted for the task of structure-preserving 3D editing, performing on par with specialized methods.

### 2.3 LATENT SPACE REGULARIZATION

To achieve the structural alignment between the generated objects, we embed these objects into a common latent space together with the transition trajectories between them. We draw inspiration from works on generative modeling of 2D images that show that alignment, disentanglement, and quality of the generated samples can be improved with regularization of trajectories between them. For example, Berthelot* et al. (2019) and Sainburg et al. (2018) directly optimize the quality of the interpolated samples with adversarial training, and StyleGAN (Karras et al., 2020) explicitly

regularizes the smoothness of the trajectories by calculating the perceptual path distance in the VGG feature space. Similarly, we employ a diffusion model as a critic that encourages plausibility of the samples on the trajectories via SDS, leading to smooth transitions and aligned objects.

## 3 PRELIMINARIES

### 3.1 NEURAL RADIANCE FIELDS

Neural radiance field (NeRF) (Mildenhall et al., 2020) is a differentiable volume rendering approach that represents the scene as a radiance function parameterized with a neural network. This network maps a 3D point $\boldsymbol{\mu} \in \mathbb{R}^3$ and a view direction $\mathbf{d} \in \mathbb{S}^2$ into a volumetric density $\tau \in \mathbb{R}^+$ and a view-dependent emitted radiance $\mathbf{c} \in \mathbb{R}^3$ at that spatial location. To render an image, NeRF queries 5D coordinates $(\boldsymbol{\mu}, \mathbf{d})$ along camera rays and gathers the output colors and densities using volumetric rendering. The ray color $\mathbf{C}$ is calculated numerically through quadrature approximation:

$$\mathbf{C} = \sum_i \alpha_i T_i \mathbf{c}_i, \qquad T_i = \prod_{j<i} 1 - \alpha_i, \qquad \alpha_i = 1 - \exp(-\tau_i \|\boldsymbol{\mu}_i - \boldsymbol{\mu}_{i+1}\|), \tag{1}$$

where $\mathbf{c}_i$ and $\tau_i$ are the radiance and density queried at the $i$'th position along the ray, and $\alpha_i$ and $T_i$ are the transmittance and accumulated transmittance.

Originally, NeRF is iteratively trained from a set of posed images. At each iteration, a batch of camera rays is randomly sampled from the set of all observed pixels and the photometric deviation between the colors $\hat{\mathbf{C}}_k$ observed along the $k$'th ray and $\mathbf{C}_k$ rendered via Equation (1) is minimized:

$$\mathcal{L}_c = \sum_k \|\mathbf{C}_k - \hat{\mathbf{C}}_k\|_2^2. \tag{2}$$

### 3.2 SCORE DISTILLATION SAMPLING

Score Distillation Sampling (SDS) (Poole et al., 2023) was proposed for fitting a NeRF to a text description of the 3D scene, without any input images, using a pre-trained text-to-image diffusion model. The NeRF is iteratively guided towards consistency with the text prompt by using the diffusion model as a critic for the rendered images. At each iteration, the image $\mathbf{x}$ is rendered for a randomly sampled camera position. A random Gaussian noise $\boldsymbol{\epsilon} \sim \mathcal{N}(\mathbf{0}, \mathbf{I})$ is added to the image and the output of the denoising diffusion model $\mathcal{E}$ is obtained via $\hat{\boldsymbol{\epsilon}} = \mathcal{E}(\mathbf{y}, t, \alpha_t \mathbf{x} + \sigma_t \boldsymbol{\epsilon})$, where $\mathbf{y}$ is the embedding of the text prompt, $t \sim \mathcal{U}(0, 1)$ is the diffusion timestep, and $\alpha_t$ and $\sigma_t$ are weighting factors. The weights $\theta$ of the NeRF network are then updated using the gradient of the SDS loss term:

$$\nabla_\theta \mathcal{L}_{\text{SDS}} = \mathbb{E}_{t, \boldsymbol{\epsilon}} \left[ w(t)(\hat{\boldsymbol{\epsilon}} - \boldsymbol{\epsilon}) \partial_\theta \mathbf{x} \right], \tag{3}$$

where $w(t)$ is another weighting factor.

Poole et al. (2023) use a NeRF-like network $F$ that maps the 3D point $\boldsymbol{\mu}$ into volumetric density $\tau$ and the diffuse RGB reflectance $\boldsymbol{\rho} \in \mathbb{R}^3$ (albedo), $i.e.$, $(\tau, \boldsymbol{\rho}) = F(\boldsymbol{\mu}; \theta)$. They obtain the emitted radiance $\mathbf{c}$ for Equation (1) via shading with a random lighting:

$$\mathbf{c} = \boldsymbol{\rho} \odot \mathbf{l}(\boldsymbol{\mu}, \mathbf{n}), \qquad \mathbf{n} = -\nabla_{\boldsymbol{\mu}} \tau / \|\nabla_{\boldsymbol{\mu}} \tau\|, \tag{4}$$

where $\mathbf{l} \in \mathbb{R}^3$ is the radiance received by the scene at the point $\boldsymbol{\mu}$ from the light sources, $\mathbf{n}$ is "surface normal", and $\odot$ is the element-wise product.

## 4 METHOD

Learning implicit 3D representations of the objects separately often produces non-aligned results (Figure 2). Our method uses a single NeRF-like network to represent a set of aligned objects together with the transitions between them. For this, we introduce a new input parameter $\mathbf{u} \in \mathbb{R}^N$ that represents a point in a latent space. We optimize the network, guiding it with a text-to-image diffusion model via SDS loss. The diffusion model is conditioned on a weighted linear combination of individual text embeddings, where the weight coefficients correspond to the elements of $\mathbf{u}$. At each iteration, we sample $\mathbf{u}$ randomly from the edges of the probability simplex. At the vertices of the simplex, the SDS loss guides the renders from the network towards consistency with the individual text prompts. At the edges of the simplex, the loss guides the renders towards image plausibility. This leads to plausible transitions between the objects in the respective pairs and, as a result, to the alignment between the objects. The overview of our method is shown in Figure 3.

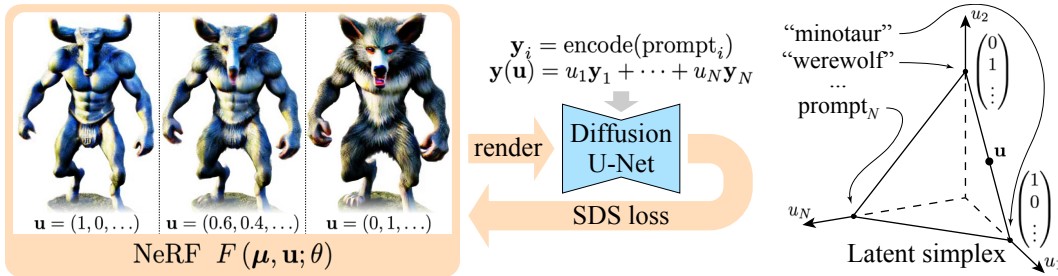

Figure 3: Overview of our method. The NeRF model, conditioned on the latent code $\mathbf{u}$ sampled from the edges of the latent simplex, produces a render. The render is passed to the diffusion model, conditioned on the linear combination of the embeddings of the text prompts. Finally, the SDS loss is backpropagated to the NeRF model.

## 4.1 GENERATION OF MULTIPLE ALIGNED 3D OBJECTS

We adopt the SDS method for the joint generation of aligned 3D objects from a set of $N$ text prompts. To do this, we embed all the objects into a common space of 3D reflectance fields represented with a single neural field. We train our network as a small generative model with the latent space built around the given set of text prompts. Specifically, we define the latent code $\mathbf{u}$ on the $(N-1)$-dimensional probability simplex $\left\{ \mathbf{u} \in \mathbb{R}^N : u_1 + \cdots + u_N = 1, u_i \geq 0 \right\}$ and assign the vertices of this simplex $\{u_i = 1\}$ to the given textual prompts. We add this latent code as an input parameter to the neural field. We train it to represent the individual 3D objects at the respective vertices of the simplex and map the linear interpolations (edges) between the latent codes at the vertices to transitions between the objects. This allows us to regularize these transitions and achieve structural alignment across the objects.

Specifically, we iteratively train the network with the SDS loss (Equation (3)). At each iteration, we sample the latent code $\mathbf{u}$ from the vertices and edges of the simplex. We render the image $\mathbf{x}$ following Equations (1) and (4), where the density and albedo now additionally depend on the latent code $(\tau, \boldsymbol{\rho}) = F(\boldsymbol{\mu}, \mathbf{u}; \theta)$. At the vertices of the simplex, we condition the diffusion model on the text embeddings of the individual prompts $\{\mathbf{y}_i\}$. At the edges, which represent transitions between the objects, we apply two kinds of regularization inspired by works on training GANs with mixed latent codes (Berthelot* et al., 2019; Karras et al., 2020).

First, we encourage the network to produce plausible 3D objects for latent codes sampled at the transition trajectories. We use the text-to-image diffusion model as a critic to evaluate and improve the plausibility through SDS. For this, we obtain the text embeddings $\mathbf{y}$ for the diffusion model as the sum of the embeddings of the individual prompts weighted with the components of the latent code $\mathbf{y}(\mathbf{u}) = u_1\mathbf{y}_1 + \cdots + u_N\mathbf{y}_N$. For the edges of the latent simplex it corresponds to linearly interpolating between the pair of embeddings of the individual text prompts. In ablation study, we show that a similar effect to a lesser extent can be achieved by conditioning the diffusion model on some general prompt independent of the objects being generated.

Second, we encourage the transitions between the objects to be smooth. We avoid doing this directly to give our model more flexibility and instead regularize the smoothness of the transitions implicitly. Specifically, we limit the depth of our neural field network, which limits the Lipschitz norm of the function parameterized by this network (Miyato et al., 2018), and enforce smoothness of rendered normal maps with a corresponding loss (see Equation (5)).

These regularization strategies encourage the network to learn a mapping from the edges of the simplex to meaningful transitions between the individual 3D objects. In our ablation study, we show that this is crucial for obtaining structurally aligned objects.

## 4.2 HYBRIDIZATION: COMBINING THE ALIGNED 3D OBJECTS

The neural network trained with our method not only represents multiple aligned 3D objects but also enables smooth interpolation of the reflectance field between these objects at each point in 3D

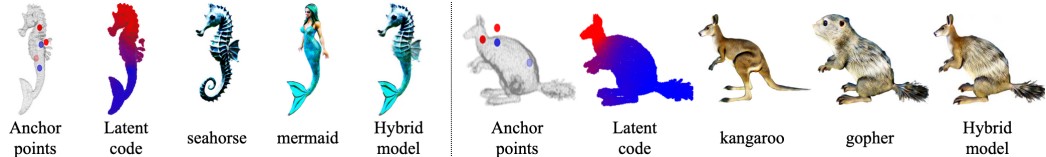

Figure 4: Our method allows us to blend different objects seamlessly. A proper alignment of multiple 3D models provides the ability to replace parts of one object with similar components of the other objects. We manually select spatial anchor points (left) and assign them to a particular model. The latent code **u** is linearly interpolated between anchors at every spatial location, resulting in a smooth distribution over 3D space (second column). The resulting objects are shown on the right.

space $\boldsymbol{\mu}$ independently. This allows for a natural and seamless fusion of objects, blending specific parts of individual generated objects into new forms, such as a gopher with a head of a kangaroo shown in Figure 4. To achieve this, the 3D space is partitioned into regions corresponding to specific objects and the reflectance field is smoothly interpolated across the boundaries of these regions. This partitioning is defined by a smooth spatial distribution of the latent code $\mathbf{u}\left(\boldsymbol{\mu}\right)$ as illustrated in the second column of Figure 4. The new hybrid model is rendered following Equations (1) and (4) with the reflectance field now depending on the spatially varying latent code $(\tau, \boldsymbol{\rho}) = F\left(\boldsymbol{\mu}, \mathbf{u}\left(\boldsymbol{\mu}\right); \theta\right)$.

### 4.3    STRUCTURE-PRESERVING TRANSFORMATION OF 3D MODELS

Our method can be easily adapted for the transformation of a given source 3D model into a target 3D model described by a text prompt while preserving the original structure, such as pose and proportions. For this, first, we set up the neural network as described in Section 4.1 for two text prompts $N = 2$. In this setup, the latent code **u** is defined on a one-dimensional segment $\{u_1 \in [0, 1]; u_2 = 1 - u_1\}$. Then, we initialize the network with the input 3D model across the whole latent space uniformly. This initialization can be done in different ways depending on the representation of the input model. In our experiments, we obtain the renderings of the input model for a random set of viewpoints and fit the network to these renderings photometrically by minimizing the loss function in Equation (2). Afterwards, we select a text prompt describing the input model (chosen manually for simplicity in our experiments) and assign the endpoints of the latent segment $u_1 = 1$ and $u_2 = 1$ to this prompt and to the target prompt, respectively. Finally, we train the network with SDS as described in Section 4.1, additionally keeping the constraint on the photometric consistency with the input model (Equation (2)) at the respective endpoint of the latent segment $u_1 = 1$.

## 5    EXPERIMENTS

We demonstrate the capabilities of our method in the three scenarios described above. We compare our method with alternatives quantitatively for the generation of pairs of aligned 3D objects and for the structure-preserving transformation of 3D models. We discuss the results of the hybridization of aligned objects generated with our method from the qualitative perspective. We implement of our method based on MVDream text-to-3D generation model (Shi et al., 2024), that uses an efficient version of NeRF Instant-NGP (Müller et al., 2022) to represent the 3D scene. In Appendix A, we show that our method is also effective in combination with a different model RichDreamer (Qiu et al., 2024) that represents 3D objects using DMTet (Shen et al., 2021). We show additional applications of our method in Appendix E. We refer the reader to the complete set of animated results of our experiments on the project page for a more complete picture. In Appendix B.4, we provide the computational costs and hardware details.

**Metrics.** We quantify three aspects of the generated pairs of aligned objects and the results of the structure-preserving transformation. The first one is the degree of alignment between the corresponding structural parts of the objects in a generated pair, or of a source 3D model and its transformed version. Measuring such alignment directly would require explicit detection of corresponding structural parts for an arbitrary pair of objects, which is a hard task by itself, even if the objects have similar structure. Recently, Tang et al. (2023) have proposed a method for finding corresponding points in pairs of images of arbitrary similar objects by matching features extracted from pretrained

Table 1: Quantitative comparison for the generation of multiple aligned 3D objects.

| | GPTEval3D, % of comparisons where our method is preferred | | | | | | | CLIP ↑ | DIFT distance ↓ % of object size |
|---|---|---|---|---|---|---|---|---|---|
| | Text-asset alignment | Text-geometry alignment | 3D plausibility | Texture details | Geometry details | Overall quality | | | |
| vs. MVEdit | 94 | 89 | 76 | 91 | 92 | 89 | MVEdit | 27.1 | 5.5 |
| vs. LucidDreamer | 61 | 77 | 70 | 56 | 80 | 77 | LucidDreamer | 26.4 | 11.3 |
| vs. GaussianEditor | 99 | 90 | 89 | 76 | 86 | 97 | GaussianEditor | 22.8 | 2.4 |
| | | | | | | | A3D (Ours) | 27.7 | 6.1 |

2D diffusion models. Based on this method, called DIFT, we define **DIFT distance** that we use to measure the structural alignment. To compute this distance for a pair of objects, we render them from the same viewpoint. We densely sample points on one of the renders and find the corresponding points on the other one with DIFT. For an ideally aligned pair of objects, a sample and its corresponding point have identical image coordinates. So, we define the DIFT distance as the average distance between these coordinates across all samples. We normalize it by the size of the objects in image space, for better interpretability. We report the value averaged across multiple viewpoints around the objects and for the points sampled for each of the objects in the pair.

Second, we measure the semantic coherence between a generated object and the respective text prompt. We measure it following the methodology of **GPTEval3D** (Mao et al., 2023), that was shown to align with human perception well. Specifically, we ask a Large Multimodal Model GPT-4o (OpenAI, 2024) to compare the 3D objects generated with two methods for the same text prompt and choose the one that is more consistent with the prompt, based on *Text-Asset Alignment* and *Text-Geometry Alignment*. We compare our method against each alternative and report the percentage of comparisons in which our method is preferred. Additionally, we measure the coherence between the generated object and the prompt using **CLIP similarity** (Jain et al., 2022), which is defined as cosine similarity between the CLIP (Radford et al., 2021) embeddings of a render of the object and the respective text prompt.

Finally, we evaluate the visual quality of the generated objects and the quality of their surface. For this, we compare the objects generated with two methods using GPTEval3D based on *3D Plausibility*, *Texture Details*, *Geometry Details*, and *Overall quality*.

## 5.1 GENERATION OF MULTIPLE ALIGNED 3D OBJECTS

We evaluate our method in the generation of sets of structurally aligned objects on 15 pairs of prompts describing pairs of objects with similar morphology but different geometry and appearance, such as a car and a carriage. We include various categories of objects, namely different kinds of animals, humanoids, plants, vehicles, furniture, and buildings, see the list of prompts in Table 5.

As no existing method targets the generation of aligned 3D objects, we adopt for comparison several methods of text-driven generation and editing of 3D models. To obtain a pair of aligned objects with such a method, we generate one of the objects from scratch and transform it into the other one. We compare with MVEdit (Chen et al., 2024a), LucidDreamer (Liang et al., 2024), and GaussianEditor (Chen et al., 2024c). MVEdit takes mesh as an input and generates multiple views of the edited object with InstructPix2Pix (Brooks et al., 2023) diffusion network. Then, the mesh is optimized photometrically to be consistent with the edited views, iteratively reducing the level of diffusion noise. LucidDreamer uses Gaussian Splatting and its own version of SDS algorithm inspired by the works on 2D image editing. GaussianEditor also uses Gaussian Splatting in combination with methods of 2D generative guidance, in particular Instruct-Pix2Pix, and additionally develops an anchor loss to control the flexibility of Gaussians. It only performs text-driven editing but not generation, so we generate the initial 3D objects for this method using MVDream (Shi et al., 2024). In our ablation study, we also compare with pairs of objects generated using MVDream independently.

We show the quantitative comparison in Table 1 and the qualitative comparison in Figure 5. Our method generates pairs of objects aligned with the text prompts and with high visual and geometric quality. It outperforms all the other methods on all evaluation criteria of GPTEval3D and w.r.t. CLIP similarity. Compared to our method, MVEdit produces less detailed objects. LucidDreamer produces noisier geometry and often struggles with multi-view inconsistency. GaussianEditor often struggles

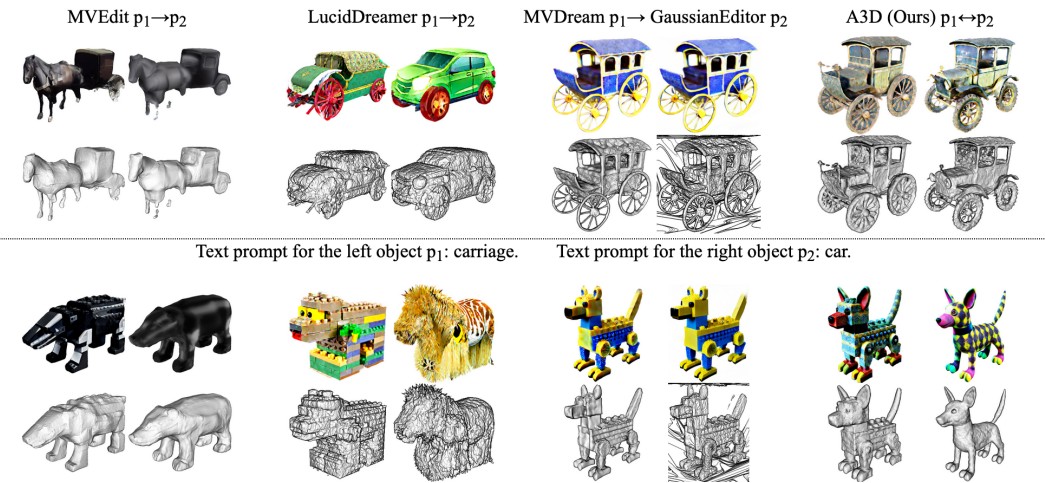

MVEdit p$_1$→p$_2$  LucidDreamer p$_1$→p$_2$  MVDream p$_1$→ GaussianEditor p$_2$  A3D (Ours) p$_1$↔p$_2$

Text prompt for the left object p$_1$: carriage.    Text prompt for the right object p$_2$: car.

Text prompt for the left object p$_1$: lego animal.    Text prompt for the right object p$_2$: animal.

Figure 5: Pairs of objects generated with existing methods and our method. The top two rows show the results for one pair of prompts written below, the bottom two rows show the results for another pair of prompts. For each object, we show a color rendering and a rendering of the geometry below it.

to obtain an object corresponding to the prompt. This may be due to Instruct-Pix2Pix (used by this method) generating inconsistent guidance from different views for synthesized objects, as it was trained on real-world data.

Pairs of objects generated with our method have a high degree of structural alignment, which is confirmed by a low DIFT distance, less than one tenth of the size of an object. W.r.t. this metric our method is only slightly outperformed by MVEdit and GaussianEditor that, while obtaining the second object in a pair by transforming the first one, often fail to change the geometry of the initial object. This leads to poor alignment with the text prompt and low quality of the generated objects overall (see the values of GPTEval3D). Moreover, the other methods, which transform one object in a pair into the other one, often generate a variant of the initial object with the geometric structure unsuitable for the other prompt, as in the *carriage-car* pair produced by MVEdit. This fundamentally limits the quality of sets of objects generated with this sequential approach. In contrast, our method jointly optimizes the set of objects so that they simultaneously share the structure and correspond to their respective text prompts well.

## 5.2 HYBRIDIZATION: COMBINING THE ALIGNED 3D OBJECTS

We show examples of the hybrid objects combining parts of aligned objects produced by our method, and illustrate the process of getting these hybrids in Figure 4. In these experiments, for better visibility we intentionally choose the hyperparameters of our method to increase the visual difference between the generated objects. To choose which part of each object we want to use, we assign several anchor points to each object and manually place these points in the common 3D space of the objects. We define the spatial distribution of the latent code $\mathbf{u}\,(\boldsymbol{\mu})$ at the location $\boldsymbol{\mu}$ (described in Section 4.2) via linear interpolation between the latent codes corresponding to the objects associated with the two closest anchors.

The examples of hybrids demonstrate that our method generates aligned 3D objects that can be seamlessly blended in different configurations. The coherent appearance of the hybrid models demonstrates a high degree of structural alignment across the generated objects. Remarkably, our method allows us to easily transition between the parts of the objects with different geometries, *e.g.*, the necks of the gopher and kangaroo, which have different diameters, or waists of the seahorse and mermaid, which have fins and hands nearby. This is in contrast to methods that represent 3D objects with a mesh (*e.g.*, MVEdit), which have to be locally adjusted first to be stitched together.

Table 2: Quantitative comparison for structure-preserving transformation.

| GPTEval3D, % of comparisons where our method is preferred | | | | | | | | |
|---|---|---|---|---|---|---|---|---|
| | Text-asset alignment | Text-geometry alignment | 3D plausibility | Texture details | Geometry details | Overall quality | | CLIP ↑ | DIFT distance ↓ % of object size |
| vs. MVEdit | 76 | 66 | 50 | 80 | 84 | 83 | MVEdit | 27.9 | 3.7 |
| vs. GaussianEditor | 94 | 94 | 83 | 97 | 97 | 100 | GaussianEditor | 24.6 | 1.8 |
| | | | | | | | A3D (Ours) | 27.8 | 7.9 |

Text prompt: beautiful princess sitting on a throne

Text prompt: skeleton of a cat

Figure 6: Objects generated with existing methods and our method from an initial 3D model on the left. Each row shows the results obtained for the text prompt below. For each object, we show a color rendering and a rendering of the geometry.

## 5.3 STRUCTURE-PRESERVING TRANSFORMATION OF 3D MODELS

We evaluate the capability of our method to transform an initial 3D model while preserving its structure on 26 text prompts. For each prompt we find a coarse initial model with the desired structure on the web, or use the SMPL parametric human body model (Loper et al., 2023) in a desired pose. In this way, we obtain, for example, a skeleton from a 3D model of a cat, or a princess on a throne from a simple model of a sitting woman, see the list of text prompts in Table 6. We compare with the same text-driven 3D editing methods as in the generation of pairs of objects.

We show the quantitative comparison in Table 2 and the qualitative comparison in Figure 6. LucidDreamer diverged for half of the scenes, so we only compare with it qualitatively. Our method generates objects aligned with the text prompts and with high visual and geometric quality, while preserving the geometric structure of the initial 3D model in terms of pose and proportions. It generally produces results on par with state-of-art specialized text-driven 3D editing methods, which is confirmed by the metrics. Our method consistently outperforms MVEdit w.r.t. the asset quality and alignment with the prompt, by producing more detailed results. Unlike MVEdit, which is restricted to superficial deformations of the surface, our method is able to add or remove significant parts of the object requested by the prompt, *e.g.*, adding the throne and crown for the princess, or shrinking the cat down to its skeleton. This also explains a slightly higher DIFT distance for our method, since these additional parts do not have the corresponding parts in the initial 3D model. LucidDreamer produces the objects with inconsistent and distorted geometry. It rarely preserves the pose and overall structure of the initial 3D model and often struggles with the Janus problem, producing the objects with multiple faces, limbs, etc. On the other hand, it generates more detailed visual appearance compared with our method.

## 6 ABLATION

We compare our method with two branches of baselines for generating pairs of objects. We refer to these baselines as (A), (B), (C), (E), (F), and to our complete method as (D).

Table 3: Quantitative ablation study.

| DIFT distance ↓, percentage of object size | | | |
|---|---|---|---|
| (A) MVDream, independently generated objects | 30.1 | | |
| (B) MVDream + multiple objects in one network | 18.7 | (F) A3D, 3-layer MLP | 14.3 |
| (C) A3D, transition plausibility with empty prompt | 14.2 | (E) A3D, 2-layer MLP | 13.5 |
| (D) A3D, transition plausibility with blended prompt (Ours) | 6.1 | (D) A3D, 1-layer MLP (Ours) | 6.1 |

In the first branch, we study the effects of embedding a set of objects into a single neural field and the importance of regularizing plausibility of transition between them. We start with the basic version of the text-to-3D framework MVDream (Shi et al., 2024) that our method is based on (A). MVDream generates pairs of objects independently from one another. We modify MVDream to parameterize two objects with a single neural field without regularizing the transition between them (B). We also implement a version of our method that regularizes the transitions using a diffusion model conditioned on an empty text prompt (C), instead of a blending of the input prompts in our full method (D).

In the second branch of comparisons we study the importance of the smoothness of the transitions. We achieve this by following the reasoning in (Miyato et al., 2018) and limiting the depth of our neural network. Specifically, in our complete method (D), we parameterize sets of objects with a multilayer perceptron (MLP) with one hidden layer on top of a feature hash grid. We evaluate two alternative designs that use MLPs with two and three hidden layers (E, F).

We show the quantitative comparison in Table 3, the qualitative comparison in Figure 8, and provide more details in Appendix D.1. Our results fully support the motivation behind the components of our method. MVDream (A) produces pairs of objects with different poses and proportions, which is confirmed by a high DIFT distance, corresponding to the nearly one third of the size of an object. Version (B) improves alignment across the objects but still does not lead to similar poses and proportions. These results show that independent generation of the objects with a single model, does not guarantee good structural alignment, which can be explained by the mode-seeking nature of SDS (Poole et al., 2023). Using an empty prompt to enforce plausibility of the transitions (C) consistently improves the alignment between the corresponding structural parts of the objects, both qualitatively and quantitatively. This shows that the key property of our method is achieved through regularization of plausibility of the transitions and not through interpolation between the latent codes of individual objects. The interpolation that we use in our complete method (D) additionally makes the objects similar to each other (while sacrificing their realism and making them more stylized if they are naturally notably different from each other) and further improves the structural alignment across the objects. Further tuning of the plausibility loss weight allows one to control the degree of alignment, as we discuss in Appendix D.2. Our experiments with increasing the depth of the network (D-F) show that enforcing the smoothness of the transition between the objects is essential for the proper alignment.

## 7    CONCLUSIONS

We present A3D, the first method designed to generate a collection of objects structurally aligned with each other. This is achieved by encouraging the transitions between the objects, jointly embedded into a shared latent space, to be smooth and meaningful, which is demonstrated to be an essential property for proper alignment. We show that, when applied to the generation of the structurally aligned objects our method outperforms the editing-based competitors in terms of the asset quality and text-object alignment, while keeping the geometric structural alignment on the state-of-art level. When applied to the 3D editing task, our method provides the results on par with recent methods specialized in this problem. Our method allows to compose novel objects seamlessly combining the parts from the different aligned objects in the generated collection. Our approach is limited to generating static aligned objects, and can not be applied to pose-changing tasks. In future work, we plan to extend our framework to changing poses and 4D video generation by experimenting with different regularization techniques for the transitions between objects.

ACKNOWLEDGMENTS

The authors acknowledge the use of the Skoltech supercomputer Zhores (Zacharov et al., 2019) to obtain the results presented in this paper. The research was partially supported by the funding of the Skoltech Applied AI center.

ETHICS STATEMENT

Our method is built on MVDream model, so it inherits all the problematic biases and limitations that this model may have. For example, MVDream fine-tuned the open-source Stable Diffusion 2.1 model (Rombach & Esser) on the Objaverse (Deitke et al., 2023) and LAION (Schuhmann et al., 2022) datasets. The LAION-400M subset of the full LAION-5B was found to contain unwanted images (Birhane et al., 2021), including inappropriate and abusive depictions. Our method may have the potential to displace creative workers through automation and increase accessibility for the creative and gaming industries. There is the risk that our method could be used to produce fake content.

REPRODUCIBILITY STATEMENT

We build our method based on the official MVDream implementation using threestudio framework (Shi et al., 2023c). We discuss the details of implementation of our method in Appendix B.1, the details of experiments with the methods that we compare with in Appendix B.3, and describe the evaluation details in Appendix C.

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

Table 4: Quantitative evaluation of the RichDreamer-based implementation of our method for the generation of multiple aligned 3D objects.

| | Text-asset alignment | Text-geometry alignment | 3D plausibility | Texture details | Geometry details | Overall quality | | CLIP ↑ | DIFT distance ↓ % of object size |
|---|---|---|---|---|---|---|---|---|---|
| vs. MVEdit | 86 | 81 | 72 | 80 | 81 | 82 | MVEdit | 27.1 | 5.5 |
| vs. LucidDreamer | 51 | 59 | 66 | 46 | 50 | 53 | LucidDreamer | 26.4 | 11.3 |
| vs. GaussianEditor | 88 | 66 | 71 | 71 | 65 | 72 | GaussianEditor | 22.8 | 2.4 |
| GPTEval3D, % of comparisons where the RichDreamer-based version of our method is preferred | | | | | | | A3D-MVDream (Ours, main) | 27.7 | 6.1 |
| | | | | | | | A3D-RichDreamer (Ours) | 28.8 | 5.8 |

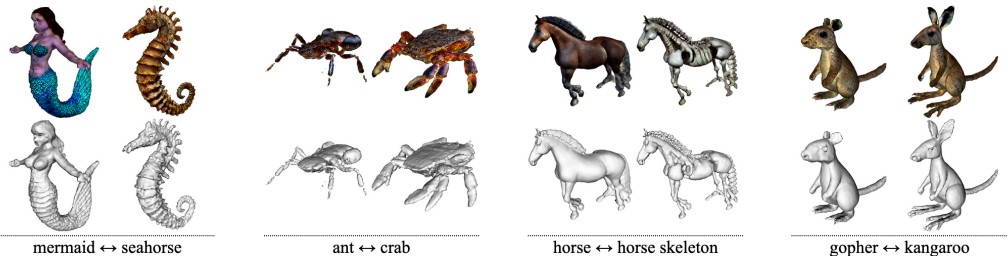

mermaid ↔ seahorse  ant ↔ crab  horse ↔ horse skeleton  gopher ↔ kangaroo

Figure 7: Pairs of objects generated using the implementation of our method based on Rich-Dreamer (Qiu et al., 2024). The respective pairs of prompts are shown below. For each object, we show a color rendering and a rendering of the geometry below it.

## A  GENERATION OF MULTIPLE ALIGNED 3D OBJECTS WITH RICHDREAMER

To evaluate the generalizability of our approach, we performed additional experiments on the generation of pairs of aligned objects using an implementation of our method based on the RichDreamer model (Qiu et al., 2024). RichDreamer employs a hybrid DMTet (Shen et al., 2021) representation for 3D content creation. It additionally integrates a special normal-depth diffusion model alongside the Score Distillation Sampling (SDS) method to refine the geometry. This approach allows for sharper edges, enhanced surface continuity, and a more realistic object appearance. In addition to improving geometry, RichDreamer also employs albedo diffusion process for texture learning. The optimization process is split into three stages. First, the coarse geometry is optimized to establish the basic shape of the 3D object. Then, this geometry is refined, improving the fidelity and detail of the object surface. Finally, the texture on the surface is generated.

In Figure 7 we show examples of the pairs of objects generated with this implementation of our method. In Table 4 we show the quantitative comparison of this implementation with the other methods. Our results with RichDreamer demonstrate the ability of our method to preserve the quality when switching the backbone. This version of our method produces the 3D objects with a higher degree of structural alignment compared with the main version based on the MVDream model. The reason for this may be that enforcing normal smoothness is easier with an SDF-based rendering backbone compared with a backbone based on radiance fields. On the other hand, the version of our method based on RichDreamer produces the results with a lower visual and geometric quality, w.r.t. GPTEval3D. One reason for this may be that RichDreamer sometimes struggles with generation of fine structures.

## B  IMPLEMENTATION DETAILS

### B.1  IMPLEMENTATION OF OUR METHOD BASED ON MVDREAM

We use the same NeRF architecture and the majority of hyperparameters for training as MVDream (Shi et al., 2024). The neural density field is parameterized with an MLP network with one hidden layer, built on top of a hierarchical feature hash grid with the dimension of 32. The grid has 16 levels starting from the resolution of 8 with 2 features per level. For SDS, we use the multi-view diffusion model from MVDream, and decrease the level of noise gradually during training.

To implement the transitions, we introduce an additional latent vector $\mathbf{u}$ that is concatenated with the coordinate embedding obtained from the hash grid and passed to the NeRF network. At each training iteration, we sample the latent vector $\mathbf{u}$ randomly from the latent simplex (described in Section 4.1). With the probability $1 - p$, we sample the latent vector from the vertices of the simplex, *i.e.*, optimize a single object, and with the probability $p$, we sample the latent vector from the edges of the simplex, *i.e.*, optimize a transition between two objects. We obtain our main results with $p = 0.5$. When the latent vector is sampled from an edge of the simplex, we additionally sample a scalar parameter $t$ from the uniform distribution $t \sim \mathcal{U}(0, 1)$ and obtain the latent vector $\mathbf{u}$ for the current training iteration via linear interpolation between the latent codes of the individual objects in the pair with this parameter $t$.

We employ two types of regularization on the normal maps: the orientation penalty described in (Poole et al., 2023) and the normal smoothness loss. Given the direction of the normal $N_{i,j}$ at the pixel with the indices $i, j$, the smoothness loss is defined as

$$\frac{1}{(H-1)(W-1)} \sum_{i=1, j=1}^{H-1, W-1} |N_{i,j+1} - N_{i,j}| + |N_{i+1,j} - N_{i,j}|, \tag{5}$$

where $H$ and $W$ are the dimensions of the normal map. We gradually increase the weight of the orientation penalty from 100 to 1000. We set the weight of the normal smoothness loss to the value of 10.

## B.2 IMPLEMENTATION OF OUR METHOD BASED ON RICHDREAMER

We mostly use the default configuration of RichDreamer, including the details of the architecture and the optimization process. This involves utilizing an MLP with one hidden layer and 64 neurons for prediction of the SDF and another MLP with the same structure for prediction of the albedo. Similarly to the implementation of our method based on MVDream, we employ the latent code $\mathbf{u}$ concatenated with the encoded points from a hash grid as the input to these networks. We initialize the geometry representation using a uniform sphere with a radius of one, and utilize SDS with Stable Diffusion 2.1 and a depth-normal diffusion model to improve the accuracy of depth predictions. For smoother interpolation between prompts, we incorporate normal consistency loss, and after experimentation, we found that setting the loss coefficient between 3 and 5 yields better results than the original configuration. To generate textures, we use a material system based on a diffuse and point-light setup without background. For generating albedo maps, we guide the prediction with an additional diffusion model.

## B.3 DETAILS OF TESTING THE OTHER METHODS

**MVEdit.** We used the official implementation of MVEdit (Chen et al., 2024b) with the default values of all hyperparameters except the denoising strength for text-guided 3D-to-3D pipeline. We changed this value from the default $0.7$ to $0.8$ to increase the scale of the changes that the method makes to the input 3D mesh. In our experiments, this lead to a higher quality of the results produced by MVEdit.

We obtain pairs of aligned objects with MVEdit in three steps, the first two of which follow the text-to-3D generation pipeline of MVEdit.

1. We generate the images of the first object in the pair using the Stable Diffusion 1.5 model (Lykon, 2023; Rombach et al., 2022) conditioned on the respective text prompt.

2. We generate the mesh of the first object from these images using the Zero123++ model (Shi et al., 2023b;a). This step includes extraction of object masks for the generated images, that we obtain using the Language Segment-Anything model (Medeiros, 2024; Kirillov et al., 2023).

3. We obtain the mesh of the second object in the pair by following the text-guided 3D-to-3D pipeline of MVEdit initialized with the previously generated mesh of the first object.

For the structure-preserving transformation of 3D models, we follow the text-guided 3D-to-3D pipeline of MVEdit in the straightforward way.

**LucidDreamer.** We used the official implementation of LucidDreamer (Liang et al., 2023) with the default values of all hyperparameters.

We obtain pairs of aligned objects with LucidDreamer in three steps, the first two of which follow the text-to-3D generation pipeline of LucidDreamer.

1. We generate a coarse point cloud of the first object in the pair using the Shape-E model conditioned on the respective text prompt.

2. We obtain the Gaussian Splatting representation of the first object by initializing it using the point cloud from the previous step and optimizing it using SDS conditioned on the same text prompt.

3. We obtain the Gaussian Splatting representation of the second object in the pair by initializing it with the previously generated Gaussian splats of the first object and optimizing it using SDS conditioned on the text prompt for the second object.

For the structure-preserving transformation of 3D models, we extract the point cloud from the source mesh and use this point cloud for initialization of the Gaussian Splatting in the generation pipeline of LucidDreamer.

**GaussianEditor.** We used the official implementation of GaussianEditor (Chen et al., 2023c) with the same values of all hyperparameters that the authors use for their experiment with the *bear* scene (Chen et al., 2023b).

GaussianEditor takes as input a Gaussian Splatting representation of the scene and performs the text-driven editing of this scene. For the generation of pairs of aligned objects, we obtain the input Gaussian Splatting for the first object in the pair generated using MVDream. For the structure-preserving transformation of 3D models, we obtain the input Gaussian Splatting for the source mesh. In both cases, we obtain the second object in the pair or the transformed object with the following steps.

1. We obtain the input Gaussian Splatting from 120 renders of the initial object, by following the original implementation of Gaussian Splatting (Kerbl et al., 2023) with the default parameters and 30k training iterations. We render the initial object from 120 camera positions evenly located around it and use the known camera parameters during optimization of the Gaussian Splatting.

2. We obtain the second object in the pair by running GaussianEditor with the prompt "Turn the *prompt_1* into a *prompt_2*", where the *prompt_1* and *prompt_2* describe the initial object and the second object in the pair respectively. For structure-preserving transformation we use the prompt "Turn *it* into a *target_prompt*".

We note that Gaussian Splatting does not provide an explicit representation of the 3D surface, so we derive the renderings of the surface (*e.g.*, shown on Figure 5) from the renderings of depth maps.

## B.4 COMPUTATIONAL COST

We run all experiments on a single Nvidia A100 GPU. To generate a single object, MVDream, which we use as the baseline of our method, requires 10k iterations, which corresponds approximately to 45 minutes. To generate *a pair of objects*, our method typically requires 20k iterations, which corresponds to 1.5 hours. The two main steps of our adaptation of MVEdit, namely text-driven 3D generation to obtain one of the objects in the pair and text-driven editing to obtain the other object, require 40 minutes in total. To generate a pair of objects, LucidDreamer typically requires 2 hours. With GaussianEditor, we generate an initial object in the pair using MVDream, which requires 45 minutes, and then edit the first object into the second one, which requires 15 minutes, so the total time required to generate a pair of objects is 1 hour. Overall, the running time of our method is comparable with the alternatives.

We have experimented with the generation of up to 5 aligned objects at a time using our method. We decided not to rely on knowledge sharing and used a simple linear heuristic for scaling the number of iterations. We add 10k optimization iterations (45 minutes) per object, so that the generation of 5 objects requires 50k iterations, which corresponds to 3 hours and 45 minutes. Informally, we have

noticed that sublinear scaling also produces the results of a high quality, so it would also be possible to use fewer iterations.

## C  EVALUATION DETAILS

For evaluation, we place the results of all methods into the same coordinate space. We manually rotate the results of MVEdit for better consistency with the other methods. We render the results from 120 camera positions evenly located around the object, consistent with the threestudio format (Shi et al., 2023c).

The results for editing-based competitors are split into two groups. The first group, which corresponds to the generative part of their pipeline produces "source" 3D objects. The second group consists of the objects that are obtained by feeding the objects from the first group to the corresponding 3D editing pipeline. The object and its corresponding transformation could be obtained by taking some object prompt from the first group and taking its complementary prompt from the pair from the second group.

**DIFT distance.**  Given a pair of images $I_A, I_B$ and corresponding masks $M_A, M_B$ we denote the DIFT (Tang et al., 2023) mappings from the first image to the second as $F_A$ and from the second to the first as $F_B$. We build two 2D point clouds $P^A, P^B$ by filtering regular 2D grids of points with masks $M_A, M_B$. We define the DIFT distance as following: $S_{DIFT} = \frac{1}{2N} \sum_{i=1}^{N} \frac{\|F_A(P_i^A) - P_i^A\|_2}{\sigma_{P_A}} + \frac{\|F_B(P_i^B) - P_i^B\|_2}{\sigma_{P_B}}$, where $\sigma_{P_A}$ and $\sigma_{P_B}$ are the diameters of the 2D pointclouds $P^A$ and $P^B$. We average the distance across the 120 viewpoints around the object.

**GPTEval3D.**  We follow the procedure proposed by (Mao et al., 2023) precisely, with one change. The version of the GPT model used in the original study is no longer available through OpenAI API, so we utilize a newer version GPT-4o. We compare each pair of methods based on 90 pairwise comparisons. For each comparison, we randomly sample a pair of objects produced by the two methods for the same text prompt and compose the request to the model. Each request consists of a pair of grids of renderings of the compared objects and a textual description of the questions to the model. Interestingly, we observed that the compressed version of the model, GPT-4o-mini, prefers the left result in the majority of comparisons regardless of the quality.

**CLIP.**  We use ViT-L/14 version of the CLIP model. We calculate the CLIP similarity between the RGB render of the object and the respective text prompt for each of the 120 viewpoints around the object and report the average value.

## D  ADDITIONAL RESULTS

In Table 5 we show the quantitative comparison of methods for the generation of multiple aligned 3D objects per each pair of objects. In Table 6 we show the quantitative comparison of methods for structure-preserving transformation per each pair of initial model and text prompt. We note that LucidDreamer diverged for half of the scenes.

### D.1  EXTENDED DISCUSSION OF ABLATION RESULTS

In Tables 7 and 8 we show the results of the quantitative ablation study per each pair of objects, and in Table 9 we show the comparison using GPTEval3D. In Figure 8, we show qualitative comparison for one pair of objects, and we refer the reader to the complete set of animated results of our experiments on the project page. We study our two main regularizations: encouraging the network to (1) learn plausible transitions between the objects, and (2) learn smooth transitions.

To demonstrate the effects of progressively decreasing the strength of smoothness regularization, we compare our method with a 1-layer MLP (D), with the versions with 2-layer MLP (E), and 3-layer MLP (F). Weakening the regularization (increasing the number of layers) leads to a lower degree of alignment, as confirmed by the DIFT distance, without any significant improvement of the visual and

Table 5: Quantitative comparison for the generation of multiple aligned 3D objects per each pair of objects.

| Prompt 1 | Prompt 2 | CLIP Similarity ↑ | | | | DIFT distance ↓, % of object size | | | |
|---|---|---|---|---|---|---|---|---|---|
| | | MVEdit | LucidDreamer | GaussianEditor | A3D (Ours) | MVEdit | LucidDreamer | GaussianEditor | A3D (Ours) |
| ant animal | crab animal | 29.2 | 24.8 | 19.9 | 28.0 | 5.0 | 10.7 | 1.9 | 5.9 |
| bicycle | motorcycle | 26.7 | 26.2 | 22.8 | 25.7 | 4.5 | 7.0 | 3.9 | 5.5 |
| bird animal | dinosaur animal | 24.0 | 25.0 | 21.5 | 26.8 | 6.4 | 15.7 | 1.6 | 6.8 |
| car | carriage | 21.8 | 28.9 | 22.2 | 29.1 | 3.0 | 10.5 | 1.6 | 4.4 |
| dwarf | minotaur | 27.2 | 24.4 | 17.0 | 26.7 | 6.0 | 9.3 | 0.9 | 5.3 |
| gopher animal | kangaroo animal | 25.0 | 25.6 | 21.2 | 28.7 | 5.9 | 10.2 | 3.4 | 4.3 |
| horse animal | horse skeleton | 26.7 | 24.4 | 25.9 | 29.1 | 2.1 | 4.2 | 1.9 | 7.2 |
| animal | lego animal | 25.9 | 25.4 | 26.0 | 24.3 | 5.1 | 16.7 | 4.7 | 5.5 |
| magnolia tree | sakura tree | 30.2 | 24.3 | 28.0 | 30.6 | 9.0 | 14.3 | 5.0 | 2.8 |
| space marine | ww2 soldier | 28.4 | 24.6 | 19.1 | 25.3 | 3.2 | 13.7 | 0.7 | 7.5 |
| mermaid | seahorse | 30.4 | 27.6 | 20.9 | 29.9 | 10.4 | 14.3 | 1.6 | 7.6 |
| man standing | robot standing | 30.7 | 26.8 | 24.9 | 29.9 | 3.4 | 11.7 | 3.6 | 5.8 |
| atakebune ship | modern yacht | 27.1 | 27.8 | 23.7 | 28.9 | 9.7 | 8.7 | 1.9 | 16.6 |
| gothic cathedral | hindu temple | 26.6 | 31.3 | 23.8 | 26.6 | 4.9 | 12.2 | 1.8 | 3.5 |
| chair | gothic throne, royal | 26.6 | 28.9 | 25.2 | 26.5 | 3.2 | 10.1 | 1.7 | 2.6 |

Table 6: Quantitative comparison for structure-preserving transformation per each pair of initial model and text prompt. We explain how we use the prompt describing the input model with our method in Section 4.3.

| Target text prompt | Prompt describing the input model | CLIP Similarity ↑ | | | | DIFT distance ↓, % of object size | | | |
|---|---|---|---|---|---|---|---|---|---|
| | | MVEdit | LucidDreamer | GaussianEditor | A3D (Ours) | MVEdit | LucidDreamer | GaussianEditor | A3D (Ours) |
| arab warrior | greek hoplite | 27.8 | 29.0 | 27.0 | 27.4 | 2.7 | 17.2 | 1.2 | 4.1 |
| astronaut | man | 27.1 | | 23.1 | 25.9 | 3.1 | | 1.0 | 6.8 |
| avocado chair | egg chair | 27.7 | | 26.4 | 32.5 | 6.2 | | 3.8 | 6.9 |
| skeleton of a cat | cat animal | 31.3 | | 28.6 | 31.6 | 3.1 | | 1.7 | 7.4 |
| clown, sitting | man, sitting | 32.1 | 29.6 | 28.7 | 31.6 | 2.2 | 13.7 | 1.2 | 6.5 |
| dragon | parrot | 24.3 | | 18.8 | 26.0 | 8.2 | | 1.8 | 12.0 |
| realistic baby duck bird | yellow duck toy | 28.6 | | 29.8 | 27.9 | 5.0 | | 3.2 | 8.4 |
| bearded dwarf with an axe | man | 26.7 | 27.6 | 23.8 | 30.5 | 3.4 | 13.1 | 1.8 | 5.4 |
| female elf woman sitting | female sitting | 27.7 | 32.7 | 30.8 | 33.6 | 1.6 | 16.2 | 1.7 | 4.5 |
| saturn planet with rings | globe on a stand | 24.5 | | 19.2 | 26.0 | 4.1 | | 1.2 | 11.6 |
| groot | man wearing jeans and t-shirt | 31.2 | | 22.7 | 31.3 | 3.6 | | 1.7 | 7.7 |
| man hunter holding a gun in both hands | male human | 28.8 | 22.5 | 26.3 | 25.6 | 2.3 | 19.5 | 1.3 | 3.6 |
| iron throne | antique wooden chair | 31.9 | | 24.9 | 27.5 | 5.4 | | 1.9 | 15.5 |
| jedi with lightsaber | man | 25.1 | | 21.3 | 27.7 | 2.4 | | 1.8 | 4.0 |
| female jedi with lightsaber | woman | 30.4 | 26.2 | 22.5 | 26.7 | 3.8 | 14.1 | 2.1 | 4.0 |
| female jedi with lightsaber | woman | 27.7 | 23.4 | 21.3 | 28.7 | 3.2 | 20.3 | 1.0 | 5.6 |
| highly detailed realistic lara croft | lara croft low poly | 29.3 | | 20.5 | 26.4 | 2.2 | | 1.6 | 3.7 |
| female marble statue | woman | 31.8 | | 30.4 | 27.2 | 2.3 | | 2.2 | 6.8 |
| space marine, warhammer | man | 24.4 | 25.9 | 20.3 | 28.0 | 6.5 | 16.6 | 2.1 | 13.1 |
| my little pony | horse animal | 29.0 | | 26.7 | 29.2 | 6.5 | | 1.3 | 15.5 |
| beautiful princess sitting on a throne | female sitting | 26.7 | 27.1 | 23.2 | 26.5 | 2.3 | 18.6 | 1.7 | 9.4 |
| robot | man | 25.6 | 25.4 | 23.2 | 24.5 | 2.6 | 12.9 | 2.0 | 10.6 |
| robot | man | 24.6 | 25.7 | 24.7 | 25.5 | 3.4 | 14.9 | 1.2 | 9.6 |
| robot, standing | man, standing | 25.8 | | 25.4 | 23.8 | 2.9 | | 1.5 | 4.8 |
| man wearing a black tailcoat with red tie | man wearing jeans and t-shirt | 28.7 | | 29.4 | 24.6 | 4.4 | | 1.6 | 4.6 |
| werewolf | man | 25.7 | 27.8 | 20.1 | 27.8 | 2.3 | 17.7 | 2.1 | 12.4 |

Table 7: Quantitative ablation study comparing our method (D) with the baselines per each pair of objects using DIFT distance. See descriptions of the baselines in Section 6.

| | | DIFT distance ↓, % of object size | | | | | |
|---|---|---|---|---|---|---|---|
| Prompt 1 | Prompt 2 | (A) MVDream | (B) MVDream+mult. | (C) Empty prompt | (D) Ours | (E) 2-layer MLP | (F) 3-layer MLP |
| ant animal | crab animal | 33.8 | 34.6 | 26.9 | 5.9 | 15.8 | 21.2 |
| bicycle | motorcycle | 17.9 | 7.6 | 5.1 | 5.5 | 7.9 | 9.1 |
| bird animal | dinosaur animal | 28.3 | 21.9 | 13.6 | 6.8 | 15.1 | 13.9 |
| car | carriage | 33.0 | 29.7 | 23.7 | 4.4 | 18.6 | 12.4 |
| dwarf | minotaur | 41.2 | 9.0 | 6.7 | 5.3 | 8.8 | 9.7 |
| gopher animal | kangaroo animal | 37.7 | 38.2 | 27.3 | 4.3 | 6.7 | 8.0 |
| horse animal | horse skeleton | 13.0 | 8.5 | 7.0 | 7.2 | 7.9 | 8.4 |
| animal | lego animal | 25.8 | 10.8 | 4.8 | 5.5 | 3.6 | 2.8 |
| magnolia tree | sakura tree | 21.3 | 0.9 | 1.1 | 2.8 | 0.7 | 1.1 |
| space marine | ww2 soldier | 21.7 | 10.0 | 13.0 | 7.5 | 9.1 | 17.8 |
| mermaid | seahorse | 34.1 | 35.8 | 27.4 | 7.6 | 24.0 | 17.9 |
| man standing | robot standing | 33.8 | 7.5 | 5.4 | 5.8 | 23.6 | 25.8 |
| atakebune ship | modern yacht | 39.5 | 30.9 | 15.3 | 16.6 | 28.3 | 26.8 |
| gothic cathedral | hindu temple | 35.7 | 8.0 | 6.1 | 3.5 | 8.0 | 12.7 |
| chair | gothic throne, royal | 34.4 | 27.2 | 29.7 | 2.6 | 25.0 | 26.9 |
| | Average | 30.1 | 18.7 | 14.2 | 6.1 | 13.5 | 14.3 |

Table 8: Quantitative ablation study comparing our method (D) with the baselines per each pair of objects using CLIP similarity. See descriptions of the baselines in Section 6.

| | | CLIP Similarity ↑ | | | | | |
|---|---|---|---|---|---|---|---|
| Prompt 1 | Prompt 2 | (A) MVDream | (B) MVDream+mult. | (C) Empty prompt | (D) Ours | (E) 2-layer MLP | (F) 3-layer MLP |
| ant animal | crab animal | 29.0 | 28.9 | 29.8 | 28.0 | 28.4 | 28.5 |
| bicycle | motorcycle | 27.3 | 26.0 | 26.6 | 25.7 | 26.5 | 27.0 |
| bird animal | dinosaur animal | 24.8 | 24.6 | 25.2 | 26.8 | 25.4 | 24.8 |
| car | carriage | 30.5 | 29.5 | 29.0 | 29.1 | 29.1 | 30.7 |
| dwarf | minotaur | 27.4 | 29.7 | 29.4 | 26.7 | 27.6 | 28.4 |
| gopher animal | kangaroo animal | 30.4 | 29.8 | 30.2 | 28.7 | 29.5 | 29.9 |
| horse animal | horse skeleton | 28.9 | 29.7 | 30.5 | 29.1 | 30.3 | 31.1 |
| animal | lego animal | 26.4 | 25.0 | 26.4 | 24.3 | 24.0 | 23.3 |
| magnolia tree | sakura tree | 31.7 | 28.1 | 24.9 | 30.6 | 28.9 | 29.0 |
| space marine | ww2 soldier | 27.0 | 26.5 | 28.7 | 25.3 | 26.6 | 29.3 |
| mermaid | seahorse | 29.9 | 29.3 | 28.9 | 29.9 | 29.0 | 28.8 |
| man standing | robot standing | 29.3 | 29.9 | 30.8 | 29.9 | 29.0 | 29.7 |
| atakebune ship | modern yacht | 28.5 | 27.7 | 28.8 | 28.9 | 28.4 | 27.9 |
| gothic cathedral | hindu temple | 28.6 | 29.0 | 29.7 | 26.6 | 28.6 | 31.4 |
| chair | gothic throne, royal | 27.7 | 27.5 | 27.5 | 26.5 | 28.1 | 28.4 |
| | Average | 28.5 | 28.1 | 28.4 | 27.7 | 28.0 | 28.5 |

Table 9: Quantitative ablation study comparing our method (D) with the baselines using GPTEval3D. See descriptions of the baselines in Section 6.

| | GPTEval3D, % of comparisons where our method is preferred | | | | | |
|---|---|---|---|---|---|---|
| | Text-asset alignment | Text-geometry alignment | 3D plausibility | Texture details | Geometry details | Overall quality |
| (A) MVDream, independently generated objects | 60.0 | 59.0 | 53.2 | 59.0 | 50.0 | 59.0 |
| (B) MVDream + multiple objects in one network | 38.0 | 37.1 | 38.7 | 38.7 | 35.5 | 33.9 |
| (C) A3D, transition plausibility with empty prompt | 47.0 | 44.9 | 65.6 | 54.4 | 68.8 | 68.8 |
| (E) A3D, 2-layer MLP | 60.2 | 52.1 | 36.8 | 54.3 | 42.6 | 42.6 |
| (F) A3D, 3-layer MLP | 59.9 | 51.4 | 50.0 | 53.6 | 50.0 | 51.5 |

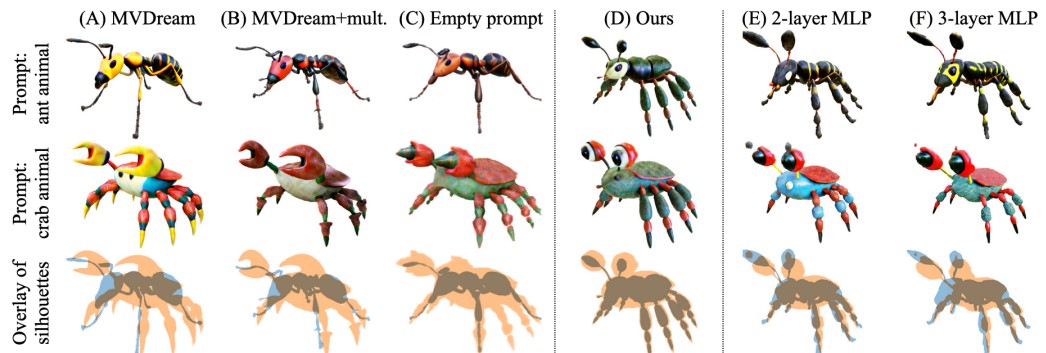

Figure 8: The first two rows show pairs of objects generated with our method (D) and the baselines (see the main text for their description). The last row shows an overlay of the silhouettes of the objects, demonstrating the alignment of their structural parts.

geometric quality, as confirmed by GPTEval3D and CLIP. We relate the preservation of the quality to the following. The Instant-NGP 3D representation, which we use for the main implementation of our method, mostly stores the neural field in the feature hash grid, while the MLP only plays an auxiliary role.

To demonstrate the effects of progressively removing the plausibility regularization, we compare our full method with the regularization conditioned on a blend of the text prompts (D), with the regularization conditioned on an empty text prompt (C), and without the plausibility regularization (B). Quantitatively, decreasing the plausibility of transitions also decreases the degree of alignment between the objects, as confirmed by the DIFT distance, but may slightly increase the visual and geometric quality of the results and their semantic coherence with the text prompts, measured with GPTEval3D and CLIP. Qualitatively, when we relax the restriction on the plausibility of transitions, the objects become less strictly aligned with each other and obtain more characteristic properties corresponding to the text prompts, especially if they are naturally different. For example, the ant and crab get generated with different numbers of legs, and the crab obtains a pair of claws; the car in the car-carriage pair changes from vintage to modern; the proportions of the gopher and kangaroo become more naturalistic.

## D.2    VARYING DEGREE OF ALIGNMENT

We achieve the alignment between the objects through regularization of transitions between them. Specifically, we encourage the network to learn plausible and smooth transitions. By varying the strength of these regularizations, we can control the degree of alignment between the objects and choose between more strict or more loose alignment.

The strength of the plausibility regularization is defined by the probability $p$ of sampling the latent code $\mathbf{u}$ from the edges of the latent simplex instead of its vertices (see Appendix B.1 for details). In Figure 9 we compare the results of our full method (D) with the results obtained with a decreased strength of the regularization (G), and without the regularization (B). When we decrease the plausibility of transitions, the objects become less strictly aligned with each other and obtain more characteristic properties corresponding to the text prompts. For example, the ant and crab get generated with different numbers of legs, and the crab obtains a pair of claws, while the proportions of the gopher and kangaroo become more naturalistic.

We demonstrate the effects of progressively decreasing the strength of smoothness regularization in our ablation study (Section 6 and Appendix D.1). We compare the results of our method with 1-layer MLP (D), 2-layer MLP (E), and 3-layer MLP (F) in Figure 8 and on the project page. The variants with more layers produce more loosely aligned objects.

On the project page, we show examples of transitions between the generated objects. The transitions are generally smooth, gradually transforming one object into another. The plausibility of the intermediate results is higher for pairs of objects with a higher degree of alignment. The implementation of our method based on RichDreamer produces smoother transitions compared to the implementation

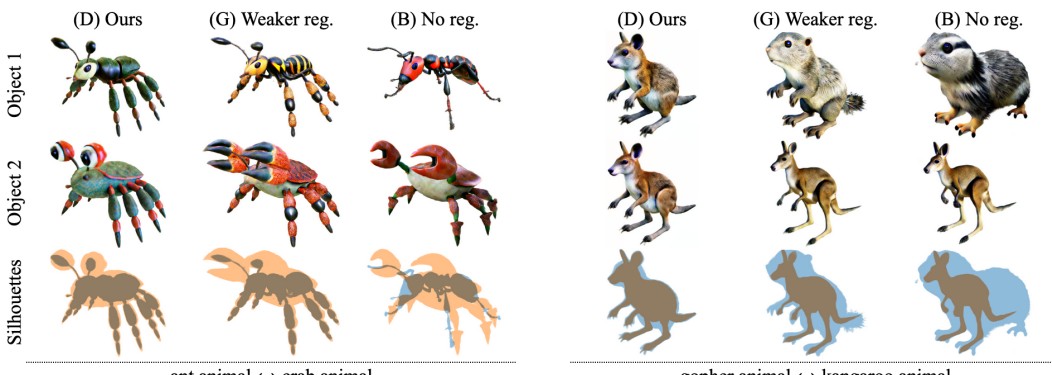

Figure 9: The first two rows show pairs of objects generated with our method in the default configuration (D), with a decreased strength of the plausibility regularization (G), and without this regularization (B). The last row shows an overlay of the silhouettes of the objects, demonstrating the alignment of their structural parts. Each three columns show the results for one pair of prompts written below.

based on MVDream, which we relate to the implicit surface prior in DMTet that encourages the network to learn smooth geometry.

# E    OTHER EXPERIMENTS

Our method can generate instances of the same object with different details. In Figure 10 we show examples of objects with adjusted accessories.

Our method can produce the results with some degree of diversity for a fixed set of text prompts, as we show at the top row of Figure 11. The diversity of the results produced by our method is mostly defined by the frameworks that we use for implementation: MVDream and RichDreamer. These frameworks use Score Distillation Sampling. Optimization with the Score Distillation Sampling employs high values of the Classifier-Free Guidance scale, which leads to mode-seeking behavior and lack of diversity. Additionally, the diffusion models used in these frameworks are tuned on the Objaverse dataset, which further decreases the diversity of the results produced by these models. One way to obtain different instances of the same set of objects with our method is to describe the desired differences in the text prompts, as we show at the bottom row of Figure 11.

Our method is robust to variations of the text prompts. In the first two rows of Figure 12 we show the results for pairs of text prompts with the same meaning but with a different phrasing. While not identical, the generated pairs exhibit a high degree of alignment between the objects and correspond to the text prompts well. In the last row of Figure 12 we show the result for a pair of prompts describing two different objects with the same attributes but using different phrasing. These objects are also aligned well and have a high quality. Overall, A3D does not require too much prompt engineering but one can expect some diversity in the results for different formulations of the text prompts.

We show a possible approach of applying our method to an image-to-3D model by performing the structure-preserving transformation of a 3D object generated with this model. We show some results obtained in this way in Figure 13. We generate the initial 3D models using the fast image-to-3D pipeline of CRM (Wang et al., 2025) and then transform these models using our method as described in Section 4.3. We use the faster implementation of our method based on the RichDreamer framework. Our method preserves the structure of the initial object generated from the image well.

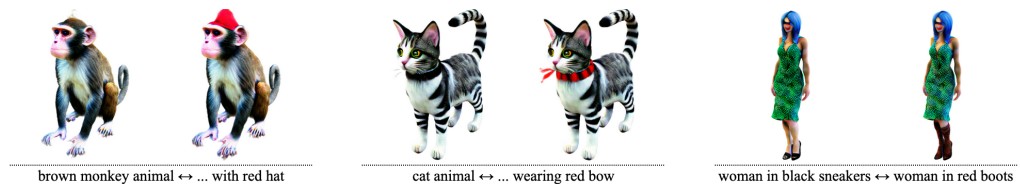

Figure 10: Pairs of instances of the same object with adjusted details generated with our method.

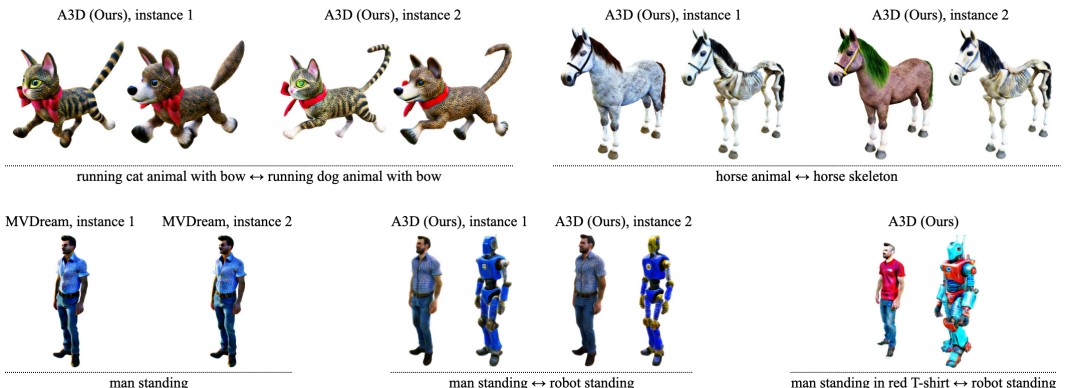

Figure 11: The top row shows two examples of slightly different instances of pairs of objects generated with our method for the fixed pairs of text prompts, written below. For some text prompts, the baseline text-to-3D framework MVDream tends to produce the exact same object every time (bottom row, left). For the respective pairs of prompts, the diversity of the results produced by our method is also limited (bottom row, middle). One way to obtain different instances of the same set of objects with our method in this case is to describe the desired differences in the text prompts (bottom row, right).

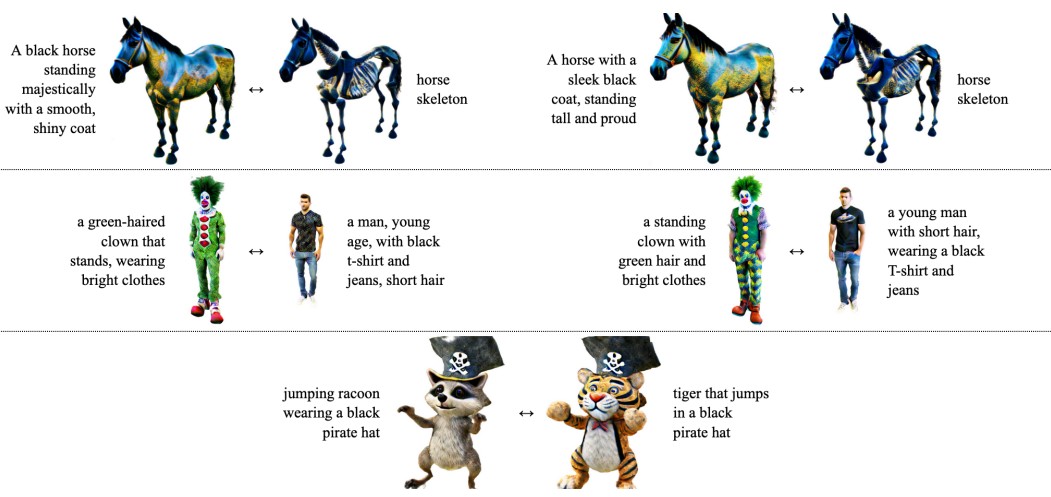

Figure 12: Each of the top two rows shows two pairs of objects generated with our method for pairs of text prompts with a similar meaning but a different phrasing. The last row shows the result for a pair of prompts describing two different objects with the same attributes but using different phrasing.

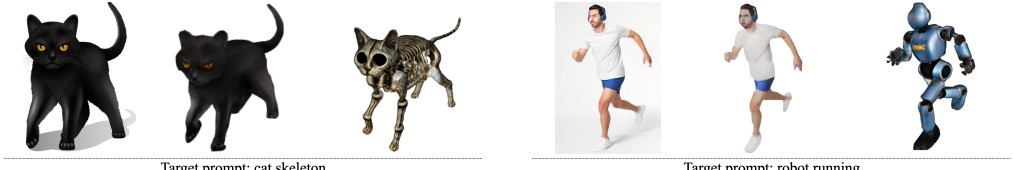

Figure 13: Examples of application of our method to an image-to-3D pipeline. Each triplet of images shows an input image, the 3D model generated with the image-to-3D pipeline, and the result of structure-preserving transformation of this model using our method with the target prompt written below. The input image of the black cat is designed by macrovector / Freepik.

