# OpenReview forum: "A3D: Does Diffusion Dream about 3D Alignment?"
_ICLR.cc/2025/Conference — ICLR 2025 Poster_

### Official Review · Reviewer_DfS8 · 2024-11-02

**Soundness:** 3
**Presentation:** 2
**Contribution:** 3
**Rating:** 6
**Confidence:** 3

**Summary:**

This paper focuses on generating a set of objects with similar structures using one text-to-3D generation model. To this aim, it introduces a  latent code to embed these desired objects into a common space. Besides training the text-to-3D generation model with their individual texts, this work enforces two kinds of regularization on the transitions between the objects. Experiments on multiple aligned 3D objects generation, combining parts of different objects into a ''hybrid'' and preserving transformations of a 3D object verify that the proposed method can generate diverse objects while maintaining structural consistency.

**Strengths:**

1. This paper introduces an interesting problem, how to make one text-to-3D generation model flexibly produce different objects while maintaining structural consistency?
2. The proposed method proposes to embed these objects into a common latent space, allowing them to share similar structures.
3. Experiments show the potential of the proposed method on different tasks, including generation of multiple objects, hybridization and structure-preserving transformation.

**Weaknesses:**

1. The paper is not well-written. It does not provide an overview to introduce the method architecture. It is better to provide a high-level flowchart of the proposed method, and then introduce the components of the proposed method by referring them to the presented flowchart. In the Method section, many details are given in the appendix.
2. The experiments lack the ablation studies for the two kinds of regularization on the transitions between objects. Can you remove the two regularizations, respectively and conduct corresponding ablation experiments to verify their effectiveness?
3. Can the proposed method be applied to image-to-3D generation models? Can you discuss the potential challenges or modifications needed to adapt their method to image-to-3D models, and how this might impact the structural consistency of the generated objects?

**Questions:**

1. I am curious if the the proposed method can be applied to image-to-3D generation methods. Since one image can be converted into a text embedding, the proposed method should support the image-to-3D setting.
2. What is the training time overhead for the proposed method?  Can you compare the training time with other methods and discuss how the training time scales with the number of objects being aligned?
3. For the ablation study, Table 3 only compares the DIFT distance metric. Can you also compare other metrics like Table 1 and Table 2?
4. I wonder if the proposed method can edit some details. For example, "a happy running man", "a sad running man", "a happy running minotaur" and "a sad running minotaur".

---

> ### Author Response · Authors · 2024-11-23
> **Response to the questions [Part 1/3]**
>
> We thank the reviewer for their valuable feedback and interest in our work.
> We address the questions and comments of the reviewer below.
>
> # Ablation study of regularizations, additional metrics
> We did perform an ablation study in the original submission, but we slightly extended it based on the suggestions of the reviewers.
> We study our two main regularizations: encouraging the network to (1) learn plausible transitions between the objects, and (2) learn smooth transitions.
> We show the qualitative comparison in Figure 6 and in the section Ablation of the supplementary anonymous webpage (https://qrd9ph4uipym.github.io/#ablation).
> We show the values of the DIFT distance, which measures the degree of alignment between the objects, in Tables 3 and 7.
> We show the scores of GPTEval3D and CLIP similarity, which measure the visual quality of the generated objects, the quality of their surface, and the semantic coherence between the generated objects and their respective text prompts, in the new Tables 8 and 9.
>
> To demonstrate the effects of progressively decreasing the strength of smoothness regularization we compare our method with a 1-layer MLP (D), with the versions with 2-layer MLP (E), and 3-layer MLP (F).
> Weakening the regularization (increasing the number of layers) leads to a lower degree of alignment, as confirmed by the DIFT distance, without any significant improvement of the visual and geometric quality, as confirmed by GPTEval3D and CLIP.
> We relate the preservation of the quality with the following.
> The Instant-NGP 3D representation [1], which we use for the main implementation of our method, mostly stores the neural field in the feature hash grid, while the MLP only plays an auxiliary role.
>
> To demonstrate the effects of progressively removing the plausibility regularization we compare our full method with the regularization conditioned on a blend of the text prompts (D), with the regularization conditioned on an empty text prompt (C), and without the plausibility regularization (B).
> Quantitatively, decreasing the plausibility of transitions also decreases the degree of alignment between the objects, as confirmed by the DIFT distance, but may slightly increase the visual and geometric quality of the results and their semantic coherence with the text prompts, measured with GPTEval3D and CLIP.
> Qualitatively, when we relax the restriction on the plausibility of transitions, the objects become less strictly aligned with each other and obtain more characteristic properties corresponding to the text prompts, especially if they are naturally different.
> For example, the ant and crab (slide 1 at https://qrd9ph4uipym.github.io/#ablation) get generated with different numbers of legs, and the crab obtains a pair of claws;
> the car in the car-carriage pair (slide 2) changes from vintage to modern;
> the proportions of the gopher and kangaroo (slide 3) become more naturalistic.
>
> Our method allows choosing the trade-off between the realism of the generated objects and alignment between them, by reducing the strength of the plausibility regularization.
> It is defined by the probability $p$ of sampling the latent code $\boldsymbol{\mathrm{u}}$ from the edges of the latent simplex instead of its vertices (see the updated Section B1 for details).
> In the new Figure 8 and the new section “Varying degree of alignment” of the supplementary webpage (https://qrd9ph4uipym.github.io/#alignment-degree), we compare the results of our full method (D) with the results obtained with a decreased strength of the regularization (G), and without the regularization (B).

---

> ### Author Response · Authors · 2024-11-23
> **Response to the questions [Part 2/3]**
>
> # Application of our method to image-to-3D generation
> Our method does not have any direct limitations that would prohibit an application in the image-to-3D scenario.
> However, a proper implementation requires a significant amount of additional work.
> The approach proposed by the reviewer, i.e., using embeddings of images instead of embeddings of text prompts, is expected to produce the objects only loosely consistent with the input images.
> CLIP embeddings of images do not provide precise information about the object and have a dimensionality different from the embeddings of text prompts, which makes them difficult to use directly with text-to-image diffusion models like Stable Diffusion.
> Image-to-3D pipelines often use various tricks to achieve high consistency between the generated object and the input image, e.g., train a special multiview-consistent sampler [2-4], or use additional 3D priors such as depth and normal maps estimated from the input image and additional model tuning [5].
> Further, image-to-3D models usually employ strong regularizations to strictly preserve the structure or pose of the object depicted in the image.
> This brings additional challenges for the generation of multiple aligned objects from a set of respective images related to possibly different structures of the objects in the images.
>
> One possible approach of applying our method to an image-to-3D model is to perform the structure-preserving transformation of a 3D object generated with this model.
> We show some results obtained in this way in the new Figure 9.
> We generate the initial 3D models using the image-to-3D pipeline of CRM [6] and then transform these models using our method as described in Section 4.3.
> We use the implementation of our method based on the RichDreamer framework [7].
> It can be initialized with the input 3D geometry directly, quicker than the MVDream-based implementation that we used to obtain our main results.
> Our method preserves the structure of the initial object generated from the image well.
>
>
> # How does the training time of A3D compare with the other methods
> To avoid any misunderstanding, we would like to clarify that we do not train any new model and that our method iteratively generates a set of objects using a pre-trained 2D diffusion model.
> Below, we compare the time required for the generation.
>
> We run all experiments on a single Nvidia A100 GPU.
> - To generate a single object, MVDream [8], which we use as the baseline of our method, requires 10k iterations, which corresponds to 45 minutes.
> - To generate _a pair of objects_, our method typically requires 20k iterations, which corresponds to 1.5 hours.
> - The two main steps of our adaptation of MVEdit [9], namely text-driven 3D generation to obtain one of the objects in the pair and text-driven editing to obtain the other object, require 40 minutes.
> - To generate a pair of objects, LucidDreamer [10] typically requires 2 hours.
> - With GaussianEditor [11], we generate an initial object in the pair using MVDream, which requires 45 minutes, and then edit the first object into the second one, which requires 15 minutes. So, the total time required to generate a pair of objects is 1 hour.
>
> Overall, the running time of our method is comparable with the alternatives.

---

> ### Author Response · Authors · 2024-11-23
> **Response to the questions [Part 3/3]**
>
> # How does the training time scale with the number of objects
> We experimented with the generation of up to 5 aligned objects at a time using our method.
> We decided to not rely on knowledge sharing and used a simple linear heuristic for scaling the number of iterations.
> We add 10k optimization iterations / 45 minutes per object, so that the generation of 5 objects requires 50k iterations, which corresponds to 3 hours and 45 minutes.
> Informally, we noticed that sublinear scaling also produces the results of a high quality, so it would also be possible to use fewer iterations.
>
>
> # Changing small details with our method
> Our method can generate instances of the same object with different details.
> In the new Figure 10, we show examples of objects with adjusted accessories.
> Adjusting facial expressions, as suggested by the reviewer, is very challenging, since the text-to-3D frameworks that we use to implement our method [7, 8] do not provide the possibility to generate high-quality human face details.
>
>
> # References
> - [1] Instant Neural Graphics Primitives with a Multiresolution Hash Encoding. SIGGRAPH 2022. https://arxiv.org/abs/2201.05989
> - [2] Zero-1-to-3: Zero-shot One Image to 3D Object. ICCV 2023. https://arxiv.org/abs/2303.11328
> - [3] SV3D: Novel Multi-view Synthesis and 3D Generation from a Single Image using Latent Video Diffusion. ECCV 2024. https://arxiv.org/abs/2403.12008
> - [4] GECO: Generative Image-to-3D within a SECOnd. Preprint. https://arxiv.org/abs/2405.20327
> - [5] RealFusion: 360° Reconstruction of Any Object from a Single Image. CVPR 2023. https://arxiv.org/abs/2302.10663
> - [6] CRM: Single Image to 3D Textured Mesh with Convolutional Reconstruction Model. ECCV 2024. https://arxiv.org/abs/2403.05034
> - [7] RichDreamer: A Generalizable Normal-Depth Diffusion Model for Detail Richness in Text-to-3D. CVPR 2024. https://arxiv.org/abs/2311.16918
> - [8] MVDream: Multi-view Diffusion for 3D Generation. ICLR 2024. https://openreview.net/forum?id=FUgrjq2pbB
> - [9] Generic 3D Diffusion Adapter Using Controlled Multi-View Editing. Preprint. https://arxiv.org/abs/2403.12032
> - [10] LucidDreamer: Towards High-Fidelity Text-to-3D Generation via Interval Score Matching. CVPR 2024. https://arxiv.org/abs/2311.11284
> - [11] GaussianEditor: Swift and Controllable 3D Editing with Gaussian Splatting. CVPR 2024. https://arxiv.org/abs/2311.14521
>
>
> # Related changes of the submission
> Below, we summarize our update of the submission related to the comments of the reviewer.
> The significant edits in the draft are highlighted in blue.
>
> 1. We added the other metrics for the ablation study in the new Tables 8 and 9 and provided an extended discussion of the ablation results in Section D1.
> 2. We added a different small ablation study in the new Section E and the respective results in the new Figure 8 and in the supplementary material.
> 3. We made a slight overall revision of the implementation details of our method for clarity in Section B1.
> 4. We added examples of application of our method to an image-to-3D model in the new Figure 9.
> 5. We added the discussion of running times of different methods in the new Section B4.
> 6. We added examples of localized edits performed by our method in the new Figure 10.

---

> ### Author Response · Authors · 2024-11-27
> **Flowchart of the method**
>
> Thank you once again for your feedback.
> We have made an additional update to the submission and added the diagram illustrating our method.
> Currently, we have added the diagram in Figure 13, in the Appendix, to avoid confusion with the numbers of the other figures.
> We will move the diagram to the main text in the final revision of the paper.
>
> Please, let us know if our response does not fully address your concerns, or if it would benefit from further clarification, or if you have additional questions or concerns.
> We will be happy to discuss.

---

> > ### Comment · Reviewer_DfS8 · 2024-12-02
> >
> > Thank you very much for your detailed response, my concerns have been addressed. It will be better to present more technical details in the Method section in the revised version. Anyway, I think this work presents an interesting problem and an effective solution for  structure-consistent text-to-object generation. Therefore, I will raise my score!

---

### Official Review · Reviewer_nGyS · 2024-11-03

**Soundness:** 3
**Presentation:** 2
**Contribution:** 3
**Rating:** 6
**Confidence:** 4

**Summary:**

The paper proposes a method that enables the joint generation of a collection of 3D objects with semantically corresponding parts aligned across them.  A3D addresses 3D editing and consistent generation by embedding the 3D objects and the transitions between them into a shared latent space and enforcing the smoothness and plausibility of these transitions. This allows for several practical scenarios that benefit from alignment between the generated objects, including 3D editing, object hybridization, and structure-preserving transformations.

**Strengths:**

1. Proposed A3D can jointly generate a collection of 3D objects aligned in structure from a set of text prompts. This method can be applied to various scenarios, such as 3D editing, object mixing, and structure-preserving transformation.
2. Compared to existing methods that generate 3D objects independently, A3D can generate a collection of 3D objects that are structurally consistent, text-aligned, and of high visual quality.
3. A3D can be used in conjunction with different text-to-3D generation frameworks to improve the generation quality while maintaining the structure.
4. The results are interesting.

**Weaknesses:**

1. Lacking a diagram that can illustrate the process of the transition and latent coding process. Only the text description on this part might be not clear enough.

2. The generated results are confined to the formed shape in the latent space, which might result in unnatural and unpleasant results. e.g., in Fig. 6, the crab and ant share the same latent shape but the generated assets are not reasonable enough for both(Too many legs for the ant and no claws for the crab).

**Questions:**

1. Can it be generalized to other diffusion models like Stable Diffusion-based SDS?

2. Is it capable of performing on Gaussian splatting-based paradigms?

3. Can this method generate diverse sets of 3D results given one single set of text prompts?

---

> ### Author Response · Authors · 2024-11-23
> **Response to the questions [Part 1/2]**
>
> We thank the reviewer for their valuable feedback and interest in our work.
> We address the questions and comments of the reviewer below.
>
> # On unnatural results, and the trade-off between realism and alignment
> There are two main reasons why some objects generated with our method look unnatural.
> The first reason is that for some objects the MVDream framework [1], which we use for the main implementation of our method, tends to produce unnatural results itself.
> For example, the MVDream baseline (A) tends to generate an ant with too few legs, see Figure 6 or an animated version of this figure in the section Ablation of our supplementary anonymous webpage (https://qrd9ph4uipym.github.io/#ablation), in slide 1.
> Overall, optimization with the Score Distillation Sampling employs high values of the Classifier-Free Guidance scale, which is known to produce cartoonish results.
>
> The second reason is that our method optimizes a pair of objects (or a larger set) that on the one hand need to correspond to their respective text prompts well, but on the other hand need to be aligned with each other.
> When a pair of prompts describes naturally different objects, such as objects with different morphologies, there may be a trade-off between these two properties.
> For some prompts it may be possible to avoid this trade-off by finding unconventional versions of the objects:
> see for example the car-carriage pair (slide 2) where our method (D) generates a vintage car to align it with the carriage, despite the tendency of the baseline MVDream (A) to generate a modern car with this prompt.
> Otherwise, our method allows choosing the trade-off between the realism of the generated objects and the alignment between them.
>
> We achieve the alignment through regularization of transitions between the objects.
> Specifically, we encourage the network to learn plausible and smooth transitions.
> We can control the degree of alignment, by varying the strength of these regularizations.
> For example, the strength of the plausibility regularization is defined by the probability $p$ of sampling the latent code $\boldsymbol{\mathrm{u}}$ from the edges of the latent simplex instead of its vertices (see the updated Section B1 for details).
> In the new Figure 8 and in the new section “Varying degree of alignment” of the supplementary webpage (https://qrd9ph4uipym.github.io/#alignment-degree), we compare the results of our full method (D) with the results obtained with a decreased strength of the regularization (G), and without the regularization (B).
> When we decrease the plausibility of transitions, the objects become less strictly aligned with each other and obtain more characteristic properties corresponding to the text prompts.
> For example, the ant and crab get generated with different numbers of legs, and the crab obtains a pair of claws, while the proportions of the gopher and kangaroo become more naturalistic.

---

> ### Author Response · Authors · 2024-11-23
> **Response to the questions [Part 2/2]**
>
> # Application of our method to different diffusion models
> We expect our method to work with other diffusion models.
> We tested it with two different frameworks: MVDream (our main results) and RichDreamer [2] (see some results in Appendix A) that use two different versions of SD model, namely 2.1 and 1.5, and tune them in different scenarios.
> Our method proved to be effective in both cases.
>
>
> # Application of our method to other 3D representations
> We expect our method to work with any differentiable rendering backbone, including Gaussian splatting-based ones.
> We tested it with two different frameworks: MVDream that uses Instant-NGP [3] and RichDreamer that uses DMTet [4].
> Our method proved to be effective in both scenarios.
> We note that building and configuring the pipeline of our method on a new backbone still requires a large amount of additional work.
>
>
> # Diversity of generated objects
> Our method can produce the results with some degree of diversity for a fixed set of text prompts, see examples in the top row of the new Figure 11.
> The diversity of the results produced by our method is mostly defined by the frameworks that we use for implementation: MVDream and RichDreamer.
> These frameworks use Score Distillation Sampling.
> Optimization with the Score Distillation Sampling employs high values of the Classifier-Free Guidance scale, which leads to mode-seeking behavior and lack of diversity.
> Additionally, the diffusion models used in these frameworks are tuned on the Objaverse dataset [5], which further decreases the diversity of the results produced by these models.
>
> One way to obtain different instances of the same set of objects with our method is to describe the desired differences in the text prompts, see an example in the bottom row of Figure 11.
>
>
> # References
> - [1] MVDream: Multi-view Diffusion for 3D Generation. ICLR 2024. https://openreview.net/forum?id=FUgrjq2pbB
> - [2] RichDreamer: A Generalizable Normal-Depth Diffusion Model for Detail Richness in Text-to-3D. CVPR 2024. https://arxiv.org/abs/2311.16918
> - [3] Instant Neural Graphics Primitives with a Multiresolution Hash Encoding. SIGGRAPH 2022. https://arxiv.org/abs/2201.05989
> - [4] Deep Marching Tetrahedra: a Hybrid Representation for High-Resolution 3D Shape Synthesis. NeurIPS 2021. https://arxiv.org/abs/2111.04276
> - [5] Objaverse: A Universe of Annotated 3D Objects. CVPR 2023. https://arxiv.org/abs/2212.08051
>
>
> # Related changes of the submission
> Below, we summarize our update of the submission related to the comments of the reviewer.
> The significant edits in the draft are highlighted in blue.
>
> 1. We added the discussion on choosing the trade-off between the realism of the generated objects and the alignment between them using our method in the new Section E, and added the respective results in the new Figure 8 and in the supplementary material.
> 2. We made a slight overall revision of the implementation details of our method for clarity in Section B1.
> 3. We added examples demonstrating that our method can produce the results with some degree of diversity for a fixed set of text prompts in the new Figure 11.

---

> ### Author Response · Authors · 2024-11-27
> **Diagram of the method**
>
> Thank you once again for your feedback.
> We have made an additional update to the submission and added the diagram illustrating our method.
> Currently, we have added the diagram in Figure 13, in the Appendix, to avoid confusion with the numbers of the other figures.
> We will move the diagram to the main text in the final revision of the paper.
>
> Please, let us know if our response does not fully address your concerns, or if it would benefit from further clarification, or if you have additional questions or concerns.
> We will be happy to discuss.

---

> ### Comment · Reviewer_nGyS · 2024-11-29
>
> Thank you for your rebuttal, and I apologize for the late reply. The framework of this paper is interesting indeed, and most of my concerns have been addressed.
>
> I understand that the time for rebuttal is too short for constructing a pipeline from scratch again. However, I am still interested in the performance of A3D in Gaussian Splatting-based methods since they are the current trends of 3D generation and reconstruction. For a simple test, I think having a trial on LGM can be a good choice, since generating one single asset only takes less than one minute and it won't take long on the training of models.
>
> As stated above, I think this paper is interesting, and above the acceptance threshold, I have decided to retain my original rating. Wish the authors the best in their future research.

---

### Official Review · Reviewer_b2FB · 2024-11-04

**Soundness:** 3
**Presentation:** 3
**Contribution:** 3
**Rating:** 6
**Confidence:** 4

**Summary:**

This paper proposes A3D, a new method to generate structurally-aligned 3D objects from a set of text prompts. A3D can first generate multiple aligned 3D objects, then make hybrids combined of different parts from the generated objects. Another advantage is that A3D can achieve smooth structure-preserving transformation given a coarse mesh.

**Strengths:**

1. The overall idea is interesting. The authors observe that the previous 3D editing method fails to generate structurally-aligned high-quality 3D objects (e.g., either lack of structure alignment or detailed texture). A3D provides an effective way to generate aligned 3D objects by introducing additional latent code conditions.
2. The results in Figure 3 demonstrate that A3D indeed learns a smooth transition between different latent codes and can be decoupled with 3D positions by setting anchor points.
3. The authors provide sufficient experiments and comparisons with previous works and A3D has better performance in generating aligned objects.
4. The presentation is clear and easy to follow. Some visualizations are useful for understanding the core idea of learning smooth transitions.

**Weaknesses:**

1. I noticed that some generated textures from A3D are still poor. Is this caused by the limitation of SDS loss or the framework of A3D?
2. I am wondering whether A3D is sensitive to text prompts. It would be better if the author could evaluate this point. is A3D robust to various text prompts?
3. In section 4.1, A3D uses some regularization strategies to supervise the network, including limiting the depth of the neural network and enforcing the smoothness of the rendered results. which will cause performance degeneration compare with raw SDS-based optimization.

**Questions:**

See the Weaknesses.

---

> ### Author Response · Authors · 2024-11-23
> **Response to the questions [Part 1/2]**
>
> We thank the reviewer for their valuable feedback and interest in our work.
> We address the questions and comments of the reviewer below.
>
>
> # What limits the quality of the generated texture
> The quality of the texture generated with A3D is limited by the text-to-3D frameworks that we use to implement our method: MVDream [1] (our main results) and RichDreamer [2] (some results in Appendix A).
> In Figure 6 and in the section Ablation of the supplementary anonymous webpage (https://qrd9ph4uipym.github.io/#ablation) we compare the results of MVDream (A) with the results of A3D (D) qualitatively.
> In the new Table 8 we compare the visual quality of the results produced by MVDream and A3D measured with GPTEval3D.
> The texture generated with our method is not inferior in quality to the texture generated with the baseline framework.
>
> The quality of the texture generated with the baseline text-to-3D frameworks that we use is mostly limited by the 3D representation and the shading model that they employ.
> It is still challenging to obtain realistic textures in text-driven 3D generation, several recent works address this challenge to some extent, e.g., [3-5].
>
>
> # Is A3D robust to variations of text prompts
> A3D is robust to variations of the text prompts.
> In the first two rows of the new Figure 12 we show the results for pairs of text prompts with the same meaning but with a different phrasing.
> While not identical, the generated pairs exhibit a high degree of alignment between the objects and correspond to the text prompts well.
> In the last row of Figure 12 we show the result for a pair of prompts describing two different objects with the same attributes but using different phrasing.
> These objects are also aligned well and have a high quality.
> Overall, A3D does not require too much prompt engineering but one can expect some diversity in the results for different formulations of the text prompts.

---

> ### Author Response · Authors · 2024-11-23
> **Response to the questions [Part 2/2]**
>
> # Ablation study of regularizations, additional metrics
> In the original submission, we performed an ablation study of our two main regularizations: encouraging the network to (1) learn plausible transitions between the objects, and (2) learn smooth transitions.
> We slightly extended the ablation study based on the suggestions of the reviewers.
> We show the qualitative comparison in Figure 6 and in the section Ablation of the supplementary webpage (https://qrd9ph4uipym.github.io/#ablation).
> We show the values of the DIFT distance, which measures the degree of alignment between the objects, in Tables 3 and 7.
> We show the scores of GPTEval3D and CLIP similarity, which measure the visual quality of the generated objects, the quality of their surface, and the semantic coherence between the generated objects and their respective text prompts, in the new Tables 8 and 9.
>
> To demonstrate the effects of progressively decreasing the strength of smoothness regularization we compare our method with a 1-layer MLP (D), with the versions with 2-layer MLP (E), and 3-layer MLP (F).
> Weakening the regularization (increasing the number of layers) leads to a lower degree of alignment, as confirmed by the DIFT distance, without any significant improvement of the visual and geometric quality, as confirmed by GPTEval3D and CLIP.
> We relate the preservation of the quality with the following.
> The Instant-NGP 3D representation [6], which we use for the main implementation of our method, mostly stores the neural field in the feature hash grid, while the MLP only plays an auxiliary role.
>
> To demonstrate the effects of progressively removing the plausibility regularization we compare our full method with the regularization conditioned on a blend of the text prompts (D), with the regularization conditioned on an empty text prompt (C), and without the plausibility regularization (B).
> Quantitatively, decreasing the plausibility of transitions also decreases the degree of alignment between the objects, as confirmed by the DIFT distance, but may slightly increase the visual and geometric quality of the results and their semantic coherence with the text prompts, measured with GPTEval3D and CLIP.
> Qualitatively, when we relax the restriction on the plausibility of transitions, the objects become less strictly aligned with each other and obtain more characteristic properties corresponding to the text prompts, especially if they are naturally different.
> For example, the ant and crab (slide 1 at https://qrd9ph4uipym.github.io/#ablation) get generated with different numbers of legs, and the crab obtains a pair of claws;
> the car in the car-carriage pair (slide 2) changes from vintage to modern;
> the proportions of the gopher and kangaroo (slide 3) become more naturalistic.
>
> Our method allows choosing the trade-off between the realism of the generated objects and alignment between them, by reducing the strength of the plausibility regularization.
> It is defined by the probability $p$ of sampling the latent code $\boldsymbol{\mathrm{u}}$ from the edges of the latent simplex instead of its vertices (see the updated Section B1 for details).
> In the new Figure 8 and the new section “Varying degree of alignment” of the supplementary webpage (https://qrd9ph4uipym.github.io/#alignment-degree), we compare the results of our full method (D) with the results obtained with a decreased strength of the regularization (G), and without the regularization (B).
>
>
> # References
> - [1] MVDream: Multi-view Diffusion for 3D Generation. ICLR 2024. https://openreview.net/forum?id=FUgrjq2pbB
> - [2] RichDreamer: A Generalizable Normal-Depth Diffusion Model for Detail Richness in Text-to-3D. CVPR 2024. https://arxiv.org/abs/2311.16918
> - [3] HiFA: High-fidelity Text-to-3D Generation with Advanced Diffusion Guidance. ICLR 2024. https://arxiv.org/abs/2305.18766
> - [4] DreamMat: High-quality PBR Material Generation with Geometry- and Light-aware Diffusion Models. SIGGRAPH 2024. https://arxiv.org/abs/2405.17176
> - [5] DreamLCM: Towards High-Quality Text-to-3D Generation via Latent Consistency Model. ACMMM 2024. https://arxiv.org/abs/2408.02993
> - [6] Instant Neural Graphics Primitives with a Multiresolution Hash Encoding. SIGGRAPH 2022. https://arxiv.org/abs/2201.05989
>
>
> # Related changes of the submission
> Below, we summarize our update of the submission related to the comments of the reviewer.
> The significant edits in the draft are highlighted in blue.
>
> 1. We added examples demonstrating that our method is robust to variations of the text prompts in the new Figure 12.
> 2. We added the other metrics for the ablation study in the new Tables 8 and 9 and provided an extended discussion of the ablation results in Section D1.
> 3. We added a different small ablation study in the new Section E and the respective results in the new Figure 8 and in the supplementary material.
> 4. We made a slight overall revision of the implementation details of our method for clarity in Section B1.

---

> ### Author Response · Authors · 2024-11-27
> **Official comment by the authors**
>
> Thank you once again for your feedback.
> Please, let us know if our response does not fully address your concerns, or if it would benefit from further clarification, or if you have additional questions or concerns.
> We will be happy to discuss.

---

### Official Review · Reviewer_2STX · 2024-11-04

**Soundness:** 3
**Presentation:** 3
**Contribution:** 3
**Rating:** 6
**Confidence:** 4

**Summary:**

This paper tackles an interesting problem of hybridization and editing regarding geometry-aligned text-to-3D generation, in which they leverage 3D shape and transitional embedding between different prompts to ensure continuous transition within the same 3D representation. By embedding these objects into a common latent space, they achieve both seamless hybridization of different shapes as well as high-quality 3D alignment between the generated scene and template 3D shape.

**Strengths:**

- This paper is well-written and easy to follow.
- The motivation the paper is sound, tackling the task of 3D shape editing through addressing hybridization and alignment of difference 3D scenes corresponding to the given shape, which I believe is an innovative and novel solution to the task at hand.
- The technical solution this paper offers to the problem is simple and effective: training a single NeRF model to handle multiple scene from different prompts during the optimization phase, and then allowing for arbitrary translation between the different scenes with interpolation of latent vectors is a elegant approach to achieve 3D alignment and controllable 3D editing of the scene.

**Weaknesses:**

- This paper uses NeRF for seamless representation of difference scenes, and I understand design choice was to enable seamless transition between different optimized scenes within the representation space of NeRF. However, more explicit 3D representations such as Instant-NGP or 3DGS offer their own advantages, such as speed and controllability, and I believe are a viable candidates for comparison. Can this method be applied to such different forms to 3D representation? If so, how do their performance different from the current model that uses NeRF?

- How much 'deformation' from the given template shape does this method allow for? Figure 4 shows the generated shapes display variations in geometry from the given template - is there any mechanism within the paper that specifically allows for this "loose" alignment (SD operating in the latent space, perhaps)? How does 3D geometry look in the "middle states" of transitioning the entire network from one object to another?

I am very much willing to raise the score if the mentioned questions are addressed during the discussion phase.

**Questions:**

See the weakness section above.

---

> ### Author Response · Authors · 2024-11-22
> **Response to the questions [Part 1/2]**
>
> We thank the reviewer for their valuable feedback and interest in our work.
> We address the questions and comments of the reviewer below.
>
>
> # Application of A3D to other 3D representations. How does the performance differ depending on the representation
> We expect our method to work with any differentiable rendering backbone.
> We tested it with two different frameworks: MVDream [1] that uses Instant-NGP [2] (our main results), and RichDreamer [3] that uses hybrid implicit-explicit 3D representation DMTet [4] (see some results in Appendix A).
> Our method proved to be effective in both scenarios.
> When based on the RichDreamer framework and the DMTet representation, our method produces smoother geometry due to the surface prior.
> On the one hand, this slightly improves the structural alignment between the generated objects, as confirmed by a lower DIFT distance compared to the version based on MVDream (see Table 4).
> On the other hand, this slightly reduces the visual quality of the results, as confirmed by the values of GPTEval3D scores (see Table 1 for the MVDream-based version, and Table 4 for the RichDreamer-based version).
> We note that adaptation of our method to a new backbone still requires additional work on configuration.
>
>
> # What mechanism allows for "loose" alignment
> We achieve the alignment between the objects through regularization of transitions between them.
> Specifically, we encourage the network to learn plausible and smooth transitions.
> By varying the strength of these regularizations, we can control the degree of alignment between the objects and choose between more strict or more loose alignment.
>
> The strength of the plausibility regularization is defined by the probability $p$ of sampling the latent code $\boldsymbol{\mathrm{u}}$ from the edges of the latent simplex instead of its vertices (see the updated Section B1 for details).
> In the new Figure 8 and in the new section “Varying degree of alignment” of the supplementary webpage (https://qrd9ph4uipym.github.io/#alignment-degree), we compare the results of our full method (D) with the results obtained with a decreased strength of the regularization (G), and without the regularization (B).
> When we decrease the plausibility of transitions, the objects become less strictly aligned with each other and obtain more characteristic properties corresponding to the text prompts.
> For example, the ant and crab get generated with different numbers of legs, and the crab obtains a pair of claws, while the proportions of the gopher and kangaroo become more naturalistic.
>
> We also demonstrate similar effects of progressively decreasing the strength of plausibility regularization in our ablation study.
> In Figure 6 and in the section Ablation of the supplementary webpage (https://qrd9ph4uipym.github.io/#ablation), we compare the results of our full method with the regularization conditioned on a blend of the text prompts (D), with the regularization conditioned on an empty text prompt (C), and without the plausibility regularization (B).
>
> We demonstrate the effects of progressively decreasing the strength of smoothness regularization in our ablation study.
> In Figure 6 and in the section Ablation of the supplementary webpage (https://qrd9ph4uipym.github.io/#ablation), we compare the results of our method with 1-layer MLP (D), 2-layer MLP (E), and 3-layer MLP (F).
> The variants with more layers produce more loosely aligned objects.

---

> ### Author Response · Authors · 2024-11-22
> **Response to the questions [Part 2/2]**
>
> # How much deformation does our method allow for
> As we discussed above, the amount of misalignment between the generated objects allowed by our method depends on the strength of the regularizations.
> On the one hand, we have examples of naturally different objects that are deformed for a more strict alignment.
> Compare the objects generated with MVDream independently (A), and with our method (D) in the section Ablation of the supplementary webpage (https://qrd9ph4uipym.github.io/#ablation): see for example the pairs bird-dinosaur (slide 2), car-carriage (slide 2), and dwarf-minotaur (slide 3).
> On the other hand, we have examples with a looser alignment.
> In addition to the pairs ant-crab and gopher-kangaroo discussed above, see for example the mermaid and seahorse that have large-scale skeletal differences in Figure 3 or in the section Hybridization of the supplementary webpage (https://qrd9ph4uipym.github.io/#hybridization, slide 1).
>
> In the section “Varying degree of alignment” of the supplementary webpage (https://qrd9ph4uipym.github.io/#alignment-degree) we show an extreme example of a loose alignment between naturally very different objects: a chair and a human.
> Our method aligns the overall pose of the human with the structure of the chair.
>
>
> # How does 3D geometry look in the transition states between the objects
> We show examples of transitions between the generated objects in the new section Transitions of the supplementary webpage (https://qrd9ph4uipym.github.io/#transitions).
> The transitions are generally smooth, gradually transforming one object into another.
> The plausibility of the intermediate results is higher for pairs of objects with a higher degree of alignment.
> We additionally show examples of transitions for pairs generated with the version of our method based on RichDreamer.
> This version produces smoother transitions, which we relate to the implicit surface prior that encourages the network to learn smooth geometry.
>
>
> # References
> - [1] MVDream: Multi-view Diffusion for 3D Generation. ICLR 2024. https://openreview.net/forum?id=FUgrjq2pbB
> - [2] Instant Neural Graphics Primitives with a Multiresolution Hash Encoding. SIGGRAPH 2022. https://arxiv.org/abs/2201.05989
> - [3] RichDreamer: A Generalizable Normal-Depth Diffusion Model for Detail Richness in Text-to-3D. CVPR 2024. https://arxiv.org/abs/2311.16918
> - [4] Deep Marching Tetrahedra: a Hybrid Representation for High-Resolution 3D Shape Synthesis. NeurIPS 2021. https://arxiv.org/abs/2111.04276
>
>
> # Related changes of the submission
> Below, we summarize our update of the submission related to the comments of the reviewer.
> The significant edits in the draft are highlighted in blue.
>
> 1. We clarified that we implement our method based on two 3D representations in the beginning of Section 5.
> 2. We added the discussion on obtaining more strict or more loose alignment with our method, and the discussion of transitions between generated objects in a new Section E, and added the respective results in a new Figure 8 and in the supplementary material.
> 3. We made a slight overall revision of the implementation details of our method for clarity in Section B1.

---

> ### Author Response · Authors · 2024-11-27
> **Official comment by the authors**
>
> Thank you once again for your feedback.
> Please, let us know if our response does not fully address your concerns, or if it would benefit from further clarification, or if you have additional questions or concerns.
> We will be happy to discuss.

---

> > ### Comment · Reviewer_2STX · 2024-11-29
> >
> > Thank you for your rebuttal, which has addressed my questions and concerns. I believe this is a good paper. I will raise my score to borderline accept (6).

---

### Official Review · Reviewer_R6mh · 2024-11-04

**Soundness:** 3
**Presentation:** 3
**Contribution:** 2
**Rating:** 6
**Confidence:** 3

**Summary:**

This paper tries to tackle the problem of text-driven 3D generation from a geometry alignment perspective. To achieve the alignment of the corresponding parts of the generated objects, this paper tries to optimize the continuous transitions in a shared latent space. Visualization results demonstrate the effectiveness.

**Strengths:**

1. The motivation of optimization in a shared latent space for smooth transitions is reasonable.

2. This writing style is easy to follow.

**Weaknesses:**

1. Fairness issues. This work is built on MVDream, which highly relies on the pre-trained knowledge from SD2.1. This raises the question: compared with other approaches, does the effectiveness of A3D come from the method itself or the pre-trained knowledge of the SD model? Relevant experiments are needed to demonstrate it. Besides, the authors are encouraged to compare different SD versions, like SD1.5 and SDXL.
2. In Tab.2, MVEdit outperforms A3D on CLIP score and DIFT distance. The authors claim that A3D can add or remove significant parts. However, this indirectly indicates that pre-trained SD models may cause unfair knowledge leakage.
3. What is the training cost compared with baseline methods, like MVEdit, LucidDreamer, and GaussianEditor?

**Questions:**

See Weaknesses

---

> ### Author Response · Authors · 2024-11-22
> **Response to the questions**
>
> We thank the reviewer for their valuable feedback.
> We address the questions and comments of the reviewer below.
>
>
> # Does the effectiveness of A3D come from the pre-trained diffusion model?
> We note that all the other methods that we compare with also rely on pre-trained diffusion models.
> The effectiveness of our method in the generation of aligned objects does not come from the diffusion model itself.
> The purpose of the diffusion model is generation of objects consistent with a text prompt, but not necessarily aligned with each other.
> In our ablation study, we show that MVDream by itself does not generate sets of aligned objects.
> Most of the time, the objects have inconsistent pose, size, proportions, or overall structure.
> See the comparison between MVDream (A) and our method (D) in Figure 6, Table 3, and the Ablation section in the supplementary webpage (https://qrd9ph4uipym.github.io/#ablation), and additionally see Figure 2, where the results of MVDream are shown on the left and the results of our method are shown on the right.
>
>
> # Application of A3D to different diffusion models
> We tested our method with two different frameworks: MVDream [1] (our main results) and RichDreamer [2] (see some results in Appendix A) that use two different versions of SD model, namely 2.1 and 1.5, and tune them in different scenarios.
> Our method proved to be effective in both cases.
>
>
> # Why does A3D make large-scale changes in structure-preserving transformation while MVEdit does not
> The large-scale changes of the initial object, i.e., adding or removing significant parts, are not directly caused by the use of the pre-trained StableDiffusion model.
> Both A3D and MVEdit [3] rely on this model.
> The key difference is that A3D employs Score Distillation Sampling, while MVEdit uses iterative refinement via ancestral sampling.
> Making large-scale changes via ancestral sampling often results in oversmoothed 3D surfaces, because the generated multi-view images are slightly spatially inconsistent with each other.
>
>
> # How does the training time of A3D compare with the other methods
> To avoid any misunderstanding, we would like to clarify that we do not train any new model and that our method iteratively generates a set of objects using a pre-trained 2D diffusion model. Below, we compare the time required for the generation.
>
> We run all experiments on a single Nvidia A100 GPU.
> - To generate a single object, MVDream, which we use as the baseline of our method, requires 10k iterations, which corresponds to 45 minutes.
> - To generate _a pair of objects_, our method typically requires 20k iterations, which corresponds to 1.5 hours.
> - The two main steps of our adaptation of MVEdit [3], namely text-driven 3D generation to obtain one of the objects in the pair and text-driven editing to obtain the other object, require 40 minutes.
> - To generate a pair of objects, LucidDreamer [4] typically requires 2 hours.
> - With GaussianEditor [5], we generate an initial object in the pair using MVDream, which requires 45 minutes, and then edit the first object into the second one, which requires 15 minutes. So, the total time required to generate a pair of objects is 1 hour.
>
> Overall, the running time of our method is comparable with the alternatives.
>
>
> # How does the training time scales with the number of objects
> We experimented with the generation of up to 5 aligned objects at a time using our method.
> We decided to not rely on knowledge sharing and used a simple linear heuristic for scaling the number of iterations.
> We add 10k optimization iterations / 45 minutes per object, so that the generation of 5 objects requires 50k iterations, which corresponds to 3 hours and 45 minutes.
> Informally, we noticed that sublinear scaling also produces the results of a high quality, so it would also be possible to use fewer iterations.
>
>
> # References
> - [1] MVDream: Multi-view Diffusion for 3D Generation. ICLR 2024. https://openreview.net/forum?id=FUgrjq2pbB
> - [2] RichDreamer: A Generalizable Normal-Depth Diffusion Model for Detail Richness in Text-to-3D. CVPR 2024. https://arxiv.org/abs/2311.16918
> - [3] Generic 3D Diffusion Adapter Using Controlled Multi-View Editing. Preprint. https://arxiv.org/abs/2403.12032
> - [4] LucidDreamer: Towards High-Fidelity Text-to-3D Generation via Interval Score Matching. CVPR 2024. https://arxiv.org/abs/2311.11284
> - [5] GaussianEditor: Swift and Controllable 3D Editing with Gaussian Splatting. CVPR 2024. https://arxiv.org/abs/2311.14521
>
>
> # Related changes of the submission
> The significant edits in the draft are highlighted in blue.
> We added the discussion of running times of different methods in a new Section B4.

---

> ### Author Response · Authors · 2024-11-27
> **Official comment by the authors**
>
> Thank you once again for your feedback.
> Please, let us know if our response does not fully address your concerns, or if it would benefit from further clarification, or if you have additional questions or concerns.
> We will be happy to discuss.

---

> > ### Comment · Reviewer_R6mh · 2024-12-02
> >
> > Thanks to authors for their clarification. I keep my initial score.

---

### Meta-Review · Area_Chair_k5HP · 2024-12-19

**Metareview:**

In this paper, all reviewers vote for acceptance. After checking the paper, the AC acknowledges the technical novelty proposed in this paper and agrees on the acceptance.

### Pros
1. This paper is well-written and easy to follow.
2. The motivation to tackle the task of 3D shape editing through addressing hybridization and alignment of different 3D scenes corresponding to the given shape.
3. Sufficient experiments and comparisons with previous works to support the effectiveness of generating aligned objects.

Please polish the manuscript, given the comments the reviewers point out. For example,
1. Reporting the computation cost or training/inference time.
2. More applications of the proposed method to other 3D representations.
3. More ablation studies.

**Additional Comments On Reviewer Discussion:**

## Points Raised by Reviewers

1. **Framework Dependency**:
   - Heavy reliance on existing frameworks like MVDream and RichDreamer, raising questions about novelty and whether improvements come from the proposed method or pre-trained models.
   - Requests for evaluations using diverse diffusion backbones, such as Gaussian splatting-based paradigms.

2. **Transition and Alignment**:
   - Effectiveness of regularization strategies for smooth and plausible transitions.
   - Concerns over trade-offs between object realism and structural alignment, especially in hybrid or significantly different objects.

3. **Experimentation and Ablations**:
   - Limited diversity in ablation metrics; requests for a broader analysis across regularizations.
   - Questions about scalability, robustness to text prompt variations, and ability to produce diverse 3D outputs for the same prompts.

4. **Presentation and Clarity**:
   - Lack of high-level architectural diagrams and overly detailed methodology in appendices.
   - Visual results in some cases (e.g., ant-crab pair) were considered unnatural, prompting questions about framework constraints.

5. **Generalization and Applicability**:
   - Inquiries about adapting the method to image-to-3D generation scenarios.
   - Potential for using different 3D representations or paradigms.

## Author Responses and Revisions

1. **Clarifications on Novelty**:
   - Highlighted how A3D improves alignment and transitions beyond pre-trained capabilities of MVDream and RichDreamer.
   - Provided comparisons with other frameworks and ablation studies to distinguish contributions.

2. **Enhanced Metrics and Examples**:
   - Introduced new evaluation metrics, including GPTEval3D and CLIP similarity, and expanded ablation studies (Tables 8-9).
   - Added failure cases and visual comparisons in the supplementary materials.

3. **Framework Adaptability**:
   - Demonstrated A3D's effectiveness across two backbones, MVDream (Instant-NGP) and RichDreamer (DMTet), and provided insights on Gaussian splatting compatibility.

4. **Improved Presentation**:
   - Added architectural diagrams and transitioned relevant details from appendices to the main text for clarity.

5. **Diversity and Prompt Robustness**:
   - Showed examples of robustness to varied text prompts and generated diverse outputs for similar prompts in Figure 11.

6. **Trade-offs in Realism and Alignment**:
   - Provided explanations and visualizations of trade-offs influenced by regularization strength (e.g., Figure 8, hybrid examples).

The AC decides to **accept (poster)** due to the paper's substantial contributions to text-to-3D alignment and its rigorous rebuttal addressing most concerns.

---

### Decision · Program_Chairs · 2025-01-22

Accept (Poster)